# mRNA display with library of even-distribution reveals cellular interactors of influenza virus NS1

Yushen Du [1,2✉], Judd F. Hultquist[3,4,5,6], Quan Zhou[1], Anders Olson[1], Yenwen Tseng[1], Tian-hao Zhang[1,7], Mengying Hong[2], Kejun Tang[2], Liubo Chen[2], Xiangzhi Meng[1], Michael J. McGregor[3,4,5], Lei Dai[1], Danyang Gong[1], Laura Martin-Sancho[8], Sumit Chanda[8], Xinming Li[9], Steve Bensenger [1,7], Nevan J. Krogan [3,4,5] & Ren Sun [1,7✉]

A comprehensive examination of protein-protein interactions (PPIs) is fundamental for the understanding of cellular machineries. However, limitations in current methodologies often prevent the detection of PPIs with low abundance proteins. To overcome this challenge, we develop a mRNA display with library of even-distribution (md-LED) method that facilitates the detection of low abundance binders with high specificity and sensitivity. As a proof-of-principle, we apply md-LED to IAV NS1 protein. Complementary to AP-MS, md-LED enables us to validate previously described PPIs as well as to identify novel NS1 interactors. We show that interacting with FASN allows NS1 to directly regulate the synthesis of cellular fatty acids. We also use md-LED to identify a mutant of NS1, D92Y, results in a loss of interaction with CPSF1. The use of high-throughput sequencing as the readout for md-LED enables sensitive quantification of interactions, ultimately enabling massively parallel experimentation for the investigation of PPIs.

[1] Department of Molecular and Medical Pharmacology, University of California, Los Angeles, CA 90095, USA. [2] Cancer Institute, ZJU-UCLA Joint Center for Medical Education and Research, School of Medicine, Zhejiang University, Hangzhou 310058, China. [3] Department of Cellular and Molecular Pharmacology, University of California, San Francisco, San Francisco, CA 94158, USA. [4] California Institute for Quantitative Biosciences, QB3, University of California, San Francisco, San Francisco, CA 94158, USA. [5] J. David Gladstone Institutes, San Francisco, CA 94158, USA. [6] Division of Infectious Diseases, Northwestern University Feinberg School of Medicine, Chicago, IL 60611, USA. [7] Molecular Biology Institute, University of California, Los Angeles, CA 90095, USA. [8] Sanford Burnham Prebys Medical Discovery Institute, 10901 North Torrey Pines Road, La Jolla, CA 92037, USA. [9] Department of Pathology and Laboratory Medicine, David Geffen School of Medicine at UCLA, L, Los Angeles, CA 90095, USA. ✉email: lilyduyushen@g.ucla.edu; rsun@mednet.ucla.edu

Defining protein–protein interaction (PPI) networks is essential for understanding the regulation of cellular biological processes. However, the accurate detection of interactions between proteins, especially the ones of low abundance, remains a challenge. Virus–host PPIs offer a particularly complex case, as viral proteins are often multifunctional and can form extensive connections with multiple cellular proteins and protein complexes[1–5]. These physical interactions are often crucial for viral replication and pathogenesis, making them attractive targets for the generation of antiviral drugs. Influenza virus, for example, leverages PPIs to hijack and/or interfere with diverse cellular pathways, including growth, apoptosis, metabolism, and the immune response[1,5–7]. A comprehensive evaluation of viral host interactions is therefore fundamental for understanding the functional connections between cellular networks and disease pathogenesis[8].

Several large-scale studies have revealed the complicated networks targeted by influenza viral proteins[1–5,9,10]. The most commonly used method for understanding these interactions is affinity purification–mass spectrometry (AP-MS)[2,7,10]. Recent improvements in molecular protocols, mass spectrometer instrumentation, and computational methods have allowed for more sensitive and accurate detection of peptides[11]. However, several limitations remain, such as the requirement for high-quality antibodies and relatively limited detection sensitivity. In parallel with AP-MS, yeast-two-hybrid[12,13] and several additional display-based methods are also being widely used, including phage display, yeast display, ribosome display and mRNA display[14–26]. These methods effectively link each protein to its genotype, allowing for multiple rounds of selection and easy downstream analysis by deep sequencing.

Among these methods, mRNA display has several unique advantages[27–29]. This approach uses a puromycin–oligonucleotide to conjugate the nascent protein with its corresponding mRNA, establishing a direct link between the protein and its mRNA. Diverse input libraries of up to $10^{14}$ sequences can then be used to generate protein–mRNA pools for selection under various conditions. By linking the protein to its genotype, multiple rounds of selections can be performed to enrich for target sequences. The high sensitivity offered by PCR and high-throughput sequencing facilitates quantitative measurement of individual sequences, enabling the detection of corresponding proteins even at low abundance. We have recently improved the efficiency of mRNA display even further to enable single-round selection of high-affinity binders by optimizing the complexity and uniformity of the library[30,31]. These improvements offer the potential to increase the scale of experiments and allow for multiple experiments to be run in parallel.

mRNA display using cDNA libraries has been reported for identifying binders of target proteins, which can be released from beads under specific elution conditions after multiple rounds of selection[32,33]. For example, Shen et al. used EGTA to elute calmodulin-binding proteins in a $Ca^{2+}$-dependent manner[32], and Ju et al. used specific caspase to cleave target proteins[33]. These unique elution conditions significantly increase the specificity, but they cannot be broadly applied to other proteins in general. The requirement for multiple rounds of selection likewise limits the potential for parallel experiments and for scale-up as a sub-library must be made for each round of selection. Furthermore, cDNA libraries are biased toward highly expressed genes, which can be represented over 10,000-fold higher than the least expressed genes. Due to their low abundance, detection of these interactors is often noisy and less reliable. We hypothesize that a more evenly distributed input library with an unbiased representation of the proteome will facilitate efficient detection of PPIs, especially those with low abundance proteins.

Here, we present an mRNA display-based method for the detection of protein–protein interactions that uses a human exon library to achieve a more even representation across the proteome. We first apply this mRNA display with library of even-distribution (md-LED) method to the multifunctional NS1 protein of influenza A virus (IAV). In addition to md-LED, we conduct AP-MS studies on NS1 to enable direct comparison between these two methods. Our list of identified interactors includes several well-described NS1 interactors. Moreover, we also uncover several new interactions that were validated through immunoprecipitation and immunoblotting. Among these novel targets, we find that NS1 can directly bind fatty acid synthase (FASN) through its N-terminal domain. Cellular lipid synthesis is significantly upregulated upon NS1 protein expression, but unaffected by expression of other viral proteins or an NS1 mutant that can not bind FASN. In addition, we explore the use of md-LED to examine the differential cellular binders of wild-type NS1 and a point mutant, D92Y, which had been previously shown to weaken NS1's known function in disrupting the interferon response. We find that the D92Y mutant failed to engage CPSF1, likely resulting in an increased host response. In summary, md-LED is a complementary approach for the identification of PPIs, particularly for the identification of low abundance interactors.

## Results

**Establishment of a human exon library for mRNA display to detect protein interactions.** Human genomic DNA was extracted from peripheral blood mononuclear cells (PBMCs) of two independent, anonymous donors (Fig. 1a). The genomic DNA was fragmented and filtered to a size of 300–700 bp (Supplementary Fig. 1a, b). Two rounds of exon enrichment were performed to generate an exon DNA library. T7 promoter and FLAG-tag DNA sequences were added onto the 5′ and 3′ termini of each enriched exon fragment, respectively (Supplementary Fig. 1c). Following in vitro transcription and mRNA purification, a puromycin linker was ligated onto the 3′ ends of the mRNA. In vitro translation was then performed using rabbit reticulocyte lysate. As the exon enrichment step did not completely remove intronic sequences, especially those directly flanking exons, we increased the input RNA during in vitro translation to ~1500-ng ligated RNA, corresponding to ~$5 \times 10^{12}$ molecules. This allowed us to maximize coverage of each exon and potentially normalize out the impact of persistent intronic sequences and artificial motifs. An initial round of FLAG purification was used on the resultant protein library to remove untranslated mRNAs, prematurely stopped proteins, nonspecific binders, and precipitated peptides. mRNA/cDNA duplexes were then generated through reverse transcription to remove secondary structures of the mRNAs, prevent mRNA degradation during selection, and enable PCR amplification post selection. This final displayed exon library consists of nascent proteins and their corresponding mRNA/cDNA, which can be used to screen for binding to immobilized protein baits, a technique we call mRNA display with library of even-distribution (md-LED). High-throughput sequencing can then be used to monitor the frequency change of each exon fragment before and after selection.

We first evaluated the library quality post-FLAG purification using next-generation sequencing. Raw sequencing reads were mapped onto human reference genome (hg19). Exon frequencies in the biological duplicates were highly correlated ($R = 0.99$, Fig. 1b). In total, 55% of total input reads contained exon sequences, an over 70-fold enrichment of exon representation relative to the human genome (1.4%). Between our two donors, on average, we covered 66% (1,268,138/1,967,640) of exons and 92% (707,359/723,789) of coding DNA sequences (CDS).

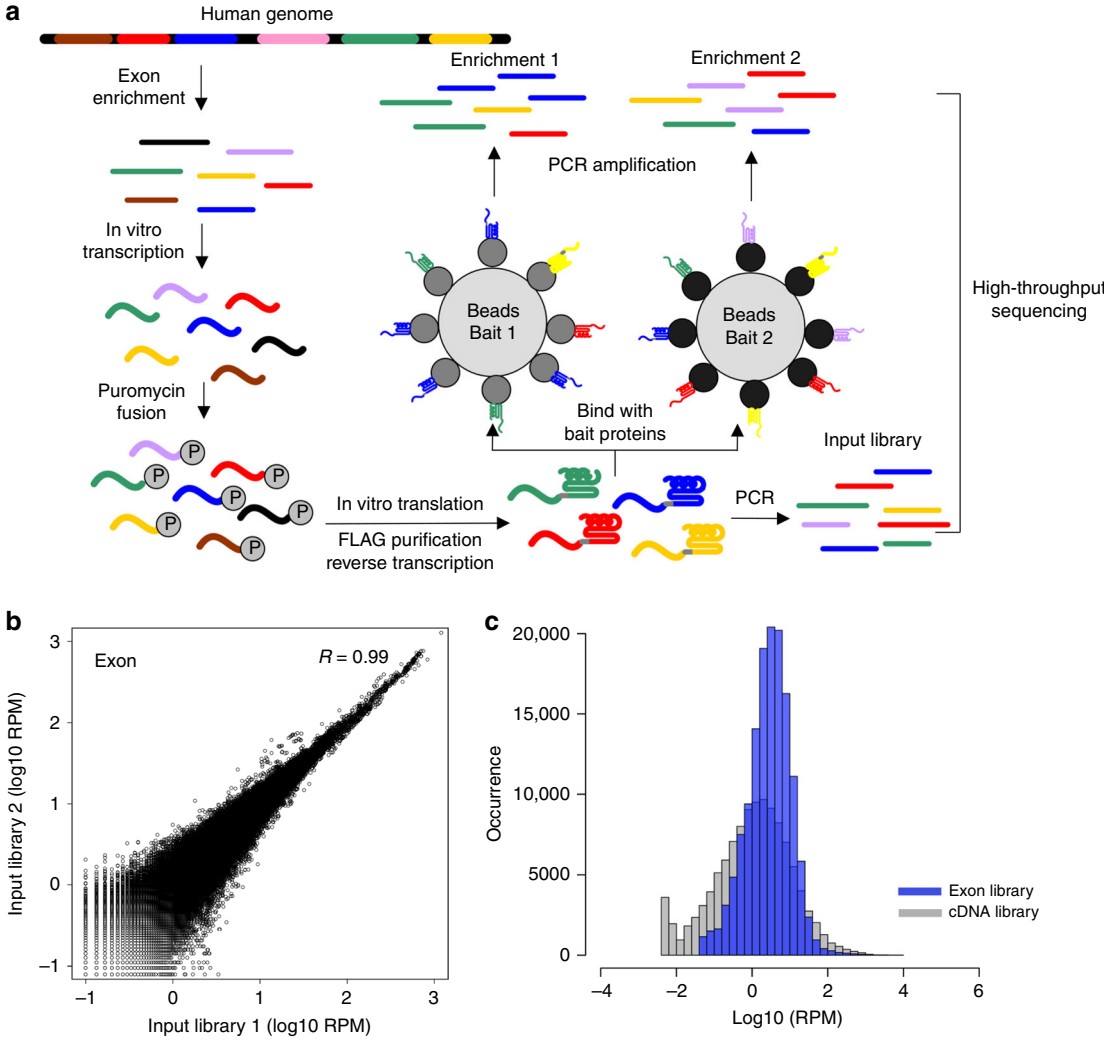

**Fig. 1 Development of md-LED to detect protein–protein interactions. a** The schematic diagram shows the experimental design of md-LED. Human exons library was enriched from fragmented DNA. The DNA fragments were transcribed in vitro and translated. Puromycin was utilized to link mRNA to its encoded protein. The nuclear acid and protein fusion complexes were pre-selected using C-terminus FLAG-tag as an input library. The pre-selected input library was then selected against bait proteins and subjected to high-throughput sequencing to determine the identity and frequency of each exon. **b** Scatter plot shows the correlation between two independent input libraries. Exon frequencies, shown as reads per million (RPM) were calculated for each replicate and strong correlation were observed with biological duplicate. **c** The distribution of transcript frequency in input libraries is shown with histogram. Blue bars represent the distribution of exon library, and gray bars correspond to cDNA library.

As an mRNA display approach using a cDNA library was previously reported for the identification of cellular binders using a multistage selection protocol[32,33], we generated a cDNA library from A549 cells to compare the coverage with our exon library. As expected, the frequency of each gene transcript in the cDNA library correlated with transcript frequency as monitored by RNA-Seq, with heavy representation bias towards highly expressed genes (Supplementary Fig. 2a). In contrast, the frequency distribution of gene transcripts is significantly more even for the exon library (two-sample KS test, $P < 0.001$), which peaked at 4–6 reads per million (RPM, Fig. 1c). We also used HTSeq software to count the gene features of the two libraries. Consistently, the exon library showed a more even distribution in the frequency of gene counts compared with the cDNA library (two-sample KS test, $p < 0.001$, Supplementary Fig. 1b). 0.4% of genes had read counts more than 1000 RPM in the cDNA library, while their representation was reduced to 0.01% in the exon library. To ensure the exon library would enable the detection of protein interactions, we performed a trial immunoprecipitation experiment to test whether the RNA–puromycin–protein prey could efficiently bind to corresponding protein antibody as the bait. We spiked a 3xHA tag sequence and an influenza NS1 sequence into the exon library to a frequency of 0.01% and synthesized a new input library of fusion proteins. Anti-HA antibody (monoclonal) and anti-NS1 antibodies (monoclonal and polyclonal) were immobilized onto protein G beads as baits, and incubated with the fusion library. The enrichment of HA or NS1 sequences after one round of selection was examined by real-time qPCR. Normalized to input, we observed 3.5–16-fold enrichment of our expected prey (Supplementary Fig. 3). Taken together, these results demonstrate the quality of our exon library and the capability of our mRNA displayed exon library to detect PPIs.

**Identification of cellular binders of influenza virus NS1 protein.** As a proof-of-principle, we employed our md-LED method to examine the cellular binders of the influenza virus (IAV) protein, NS1 (A/WSN/33 H1N1 strain). NS1 is important for efficient virus replication, being largely responsible for counteracting the host immune response and interfering with multiple

cellular pathways[6,34,35]. NS1 with a C-terminal HA tag was expressed in 293T cells, and conjugated to anti-HA beads alongside GFP-HA as a control. Extensive washing steps were performed to clean the conjugated bait proteins. These purified baits were then incubated with the fusion libraries for 3 h, precipitated, washed, and the precipitated fractions prepared for deep sequencing (Supplementary Fig. 4). Enrichment scores of individual exon and the corresponding gene transcript were calculated as the relative frequency in the output library to that in the input library. Due to the stochastic nature of the genome fragmentation, each exon fragment may contain unique flanking sequences. While the FLAG purification step is designed to remove aberrant fragments, it should be noted that some flanking sequences could impact the binding properties of exons. To reduce the effect of this possible interference, frequency of each exon was tallied independent of the flanking sequence read. The highest enrichment score for all exons mapping to each transcript was selected to represent the binding capacity of the corresponding protein to the bait (Supplementary Fig. 5).

Six biological replicates of NS1 selection were performed, revealing robust correlation ($R = 0.94$–$0.99$) for all pairwise comparisons, Supplementary Fig. 6). After filtering out nonspecific binders that were enriched in the GFP control condition, 25 proteins were identified consistently in all six replicates and considered as potential interactors with NS1 (Fig. 2a; Supplementary Data 1). Among the identified binders, four of them (CPSF4, CPSF1, PABPC1, and NFX1) were reported and validated by Immunoprecipitation (IP)-Western, and one additional factor (SF3B2) was identified through previous published AP-MS screening[1,34–38].

To validate these interactions, we performed co-immunoprecipitations (co-IPs) with each interactor as the bait (FLAG-tagged) and NS1 as the prey (2×Strep-tagged). Of the 25 candidate interactors, we were able to express 23 of them stably in 293T cells (except ADAMTS6 and DONSON, Fig. 2b). Precipitations were performed in four biological replicates alongside FLAG-tagged GFP and an empty vector as controls. In total, 14 of the 23 expressed candidates were able to co-precipitate NS1 with varying affinity (Fig. 2b). Several of these NS1 interactors have not been previously reported, including PSMB10, AKT1, FASN, DDX6, INPP5B, DNAAF3, TFAP2C, TTLL6, and USP54 (Supplementary Data 1). These interactions were also confirmed by reciprocal co-IP (Supplementary Fig. 7).

GO analysis revealed that the mRNA surveillance pathway and the regulation of lipid metabolic processes were amongst the top ranked pathways enriched in NS1 binders (Fig. 2c). We further mapped the identified binders onto the cellular gene interaction network using the STRING database (Fig. 2d)[39]. The interaction between NS1 and the mRNA cleavage and polyadenylation specificity factor (CPSF) complex has been well described and is essential for shutoff of the host response[34]. AKT1 was found to act as a hub protein with multiple connections to various complexes. The NS1 protein is known to induce the activation of the PI3K/AKT pathway, which supports viral replication, but no direct physical connection with AKT1 has been previously reported. AKT1, PDGFRA, and FASN are all involved in the regulation of cellular lipid metabolism, as we explore further below. In addition to these clusters, a few other novel interactors have known functional roles in viral replication (Fig. 2d). Overall, the identified binders were significantly enriched for influenza virus host factors when compared with cellular proteins globally (chi-square test, $P = 0.002$)[1], reflecting the linkage between physical and functional interactions.

**Comparison between md-LED and AP-MS.** To obtain a detailed comparison between md-LED and AP-MS, we performed quantitative AP-MS using the same NS1. NS1 and GFP were cloned with a C-terminal 2xStrep tag into a lentiviral vector, and the subsequent virus was used to transduce A549 cells. An antibody against the Strep tag was used to affinity purify the baits and affiliated protein complexes in three biological replicates. Samples were subjected to on-bead digest, and the resultant peptides analyzed by tandem mass spectrometry[40,41]. As NS1 is known to interact with the interferon (IFN) pathway, and the basal expression level of many IFN-stimulated genes is low in A549 cells, these experiments were performed in the presence and absence of 12-h pre-treatment with type I IFN (IFNα at 1000 U/ml). Interacting proteins identified by mass spectrometry were scored for confidence based on their specificity, reproducibility, and abundance using the MiST scoring algorithm[40,41]. A total of 316 proteins were found to interact with NS1 with a MiST score >0.8. In total, 156 baits were found regardless of treatment condition, 44 were identified only in the absence of IFN, and 116 proteins were identified only in the presence of IFN (Supplementary Data 2). Among the 25 genes that were identified with high-confidence by md-LED, DDX6 and CPSF1 were identified by both methodologies. Nevertheless, GO analysis revealed an enrichment of similar major pathways, including RNA processing and RNA 3′ processing (Fig. 3a).

One explanation for the limited overlap is that md-LED preferentially identifies direct interactors while AP-MS will pull down entire protein complexes. To examine this, we extended the md-LED database by the inclusion of known protein complexes and compared again to the AP-MS data. We extracted all validated human protein complexes from the CORUM database that included our 25 interactors, increasing the data set to 232 proteins (Supplementary Data 3)[42,43]. In this extended data set, 20 proteins overlapped directly with AP-MS result while the enriched pathways displayed extensive overlap (Fig. 3b, c; Supplementary Fig. 8). To further compare these results to published data sets, we identified 491 proteins in the literature that had been reported to interact with NS1, including data from three AP-MS studies, one yeast-two-hybrid study, and all factors from the VirHost database[1,2,7,10,44]. In total, 49 (21%) of extended md-LED binders and 155 (49%) AP-MS binders overlapped directly with these prior reports. Overall, these data suggest that while both AP-MS and md-LED can be used to identify high-confidence interactors, they have distinct advantages for the identification of protein complexes and direct binders, respectively.

One potential advantage of md-LED over AP-MS is that it enables the identification of low abundance interactors. To examine this quantitatively, we compared the cellular abundance of identified binders in our matched md-LED and AP-MS data sets. Mass spectrometry data of the A549 cellular proteome were obtained from PRIDE and used to calculate the average abundance of each set of interactors[45]. We observed a significantly lower abundance of md-LED interactors compared with AP-MS interactors (Fig. 3d). Indeed, of the 14 interactors validated by co-IP, 8 were undetected in the whole cellular proteome of A549 cells. This was further confirmed at the transcript level by RNA-Seq of A549 cells. The 25 binders identified by md-LED, including the 14 ones that were validated, had significantly lower transcript levels than those identified by AP-MS ($P = 3.2e-6$ for all md-LED hits comparing with AP-MS hits, and $P = 0.009$ for the validated hits, two-tailed Wilcoxon rank-sum test, Fig. 3d). These results emphasize the power of md-LED to detect low abundance interactors that would otherwise be missed by traditional approaches.

As the exon library was largely composed exon fragments, theoretically we should be able to extrapolate out the exact binding domains for each interacting protein. We see evidence of

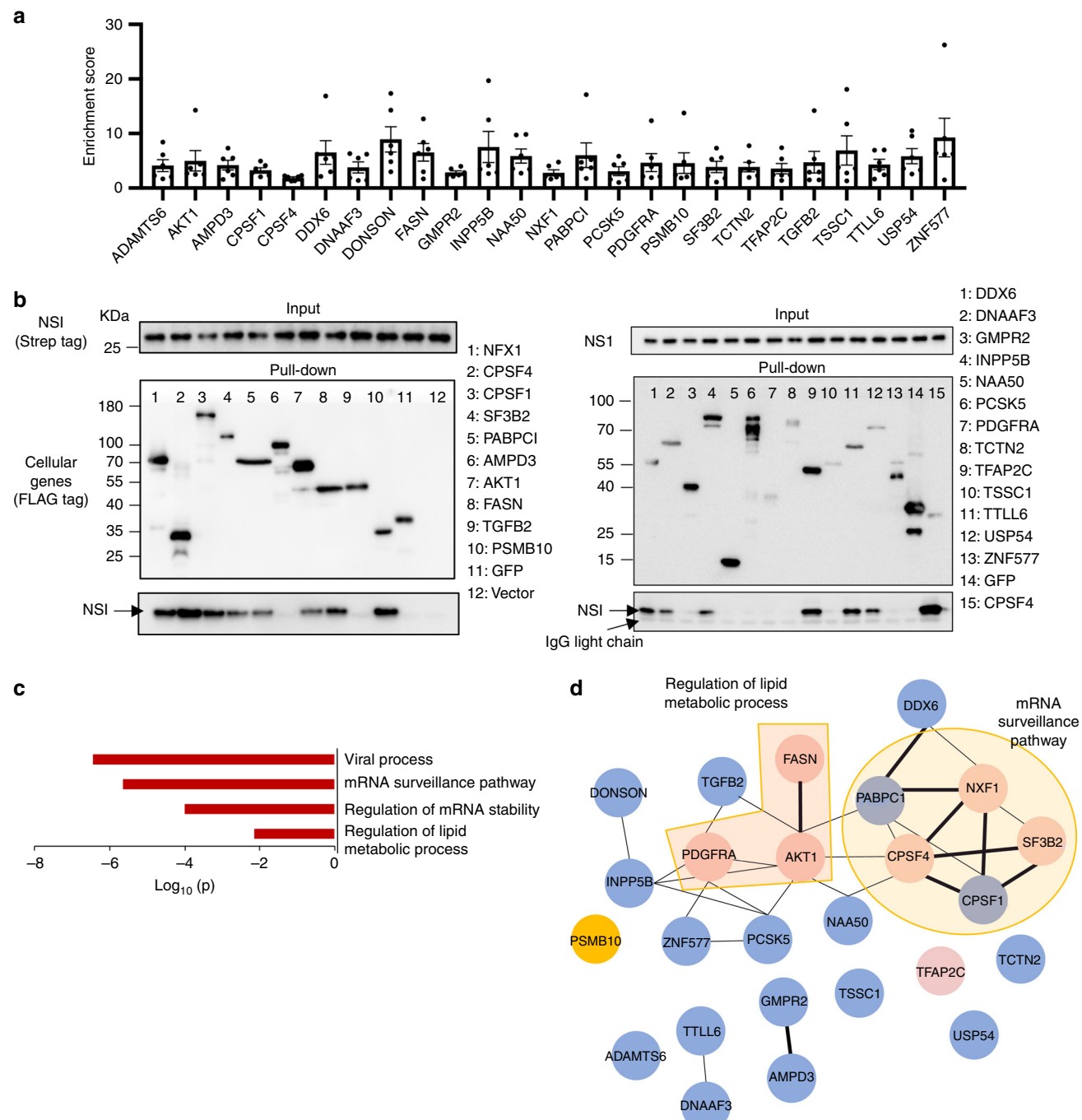

**Fig. 2 Identification of cellular interactors of influenza NS1 protein. a** Cellular proteins identified to be interacting with NS1. The corresponding enrichment score is shown ($N = 6$ biologically independent samples). Data are presented as mean values $+/-$ SEM. **b** Interactions between NS1 protein and indicated cellular binders were examined by immunoprecipitation (IP)-western. Strep-tagged NS1 protein and one of the FLAG-tagged cellular proteins were co-expressed in 293T cells. Cells were lysed 48-h post transfection. Cellular proteins were pull-down with FLAG beads and detected for co-eluted NS1 protein using antibody against Strep or NS1. Four biological replicates were performed, and a representative experiment is shown. **c** GO enrichment analysis of genes that were identified to be interacting with NS1 through md-LED. Metascape was applied for this analysis, which utilized the hypergeometric test and Benjamini–Hochberg p-value correction algorithm. **d** Cellular interaction network of NS1 binders. Interactions with confidence >0.15 in STRING database were included. Each node represents a cellular binder, and the width of edge represents the confidence level of interactions. Pink nodes are known host-dependent factors, and the orange nodes are host-restriction factors for influenza viral replication, extracted from the IAV database. The light yellow circle indicates the cluster of genes involved in mRNA surveillance. The orange shade indicates the cluster of genes in the pathway of regulation of lipid metabolic process. Source data for **a** is provided in Supplementary Data 1. Source data for **b** is provided in Source Data file.

this in the previously reported interaction domains of CPSF4 and PABPC1 (Supplementary Fig. 9). Loops 2 and 3 (the entirety of exon 3 and part of exon 4) of CPSF4 were shown to be important for NS1 binding based on a co-crystal structure (PDB: 2RHK),

and exon 3 displays the strongest enrichment score. Similarly, residues 365–535 (exons 8–11) of PABPC1 were found to be essential for the NS1 interaction based on deletion mapping[36,37], and exons 8–11 were also among the most enriched segments in

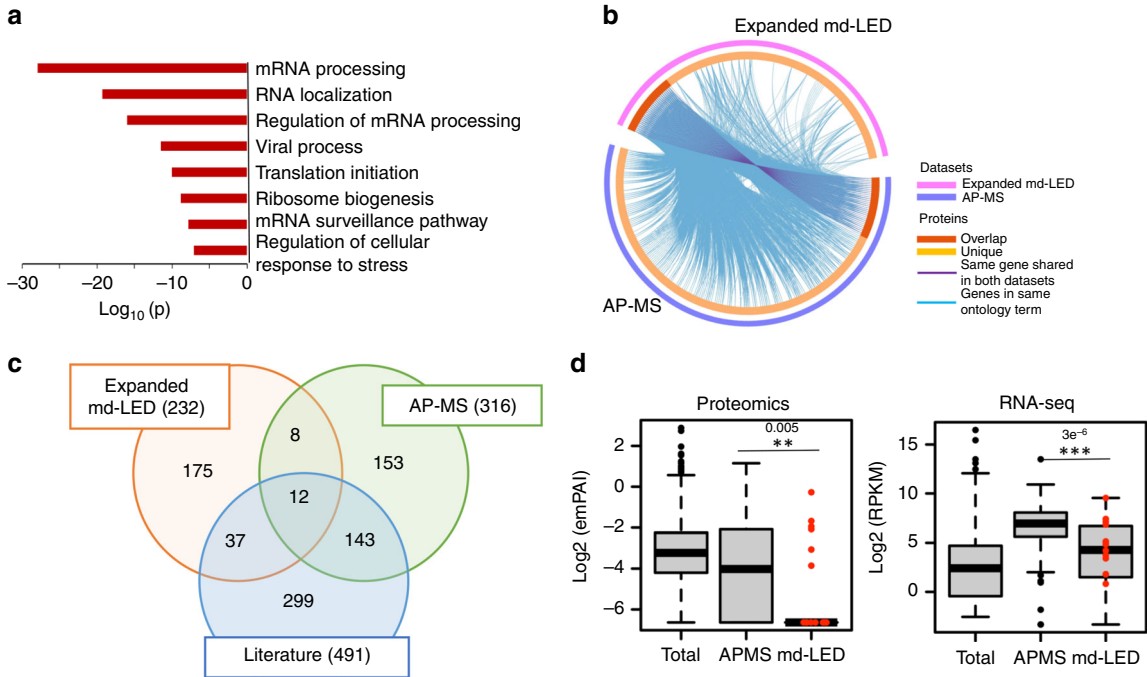

**Fig. 3 md-LED facilitates identification of binders of low abundance. a** GO enrichment analysis of genes that were identified to be interacting with NS1 through AP-MS. Metascape was applied for this analysis, which utilized the hypergeometric test and Benjamini–Hochberg P-value correction algorithm. **b** The Circos plot shows the overlap between proteins identified through expanded md-LED with AP-MS. On the outside, each arc represents the identity of gene list, using the same color code as shown on the legend. On the inside, the dark orange color represents the genes that appear in both methods, and light orange color represents genes that are unique to one method. Purple lines link the same gene that are shared by multiple gene lists. Blue lines link the different genes where they fall into the same ontology term. **c** Venn plot shows the overlap among the proteins identified through expanded md-LED, AP-MS, and literature. **d** The cellular abundance of identified protein binders were compared among total cellular proteins, md-LED, and AP-MS methods. Cellular abundance of each protein was quantified with a published cellular proteome database (PRIDE, project: PXD000418, left panel) and correlated transcript abundance was quantified through RNA-seq (right panel). The 14 cellular hits validated by Co-IP from md-LED method are marked in red. Center represents median, bounds of box represent first and third quartile, whiskers represent median $+/-1.5$ IQR. *$P < 0.05$, **$P < 0.01$, ***$P < 0.001$ (two-tailed Wilcoxon rank-sum test, the exact P-value is labeled on the figure). Source data for **b** and **c** is provided in Supplementary Data 3. Source data for **d** is provided in Supplementary Data 2 and Source Data file.

our md-LED data set (Supplementary Fig. 9). While intriguing, several limitations to this approach may prevent its use for accurate de novo identification of interacting domains, including domain disruption, high noise caused by random fragmentation, and binding competition ("Discussion").

Collectively, the above results suggest that md-LED is capable of identifying cellular interactors of proteins and in a manner complementary to AP-MS approaches. md-LED also offers the potential advantages of identifying low abundance interactors and in facilitating the identification of binding domains.

**NS1 regulates cellular lipid metabolism by binding to FASN.** Given the enriched detection of lipid regulating proteins in our md-LED data set, we focused on the interaction between FASN (fatty acid synthase) and NS1 for further functional analysis. FASN is a multifunctional protein that is critical for catalyzing and regulating fatty acid synthesis in mammalian cells[46]. While influenza virus infection significantly induces fatty acid bio-synthesis[47], the mechanisms driving this induction have remained elusive. Above, we demonstrated an interaction between influenza virus NS1 and FASN by overexpression (Fig. 2b; Supplementary Fig. 9). We confirmed an interaction with endogenous FASN in infected A549 cells using an anti-NS1 antibody to precipitate NS1 containing complexes (Fig. 4a). Upon NS1 overexpression, the mRNA expression level of *FASN* did not significantly change, but the protein steady-state level increased (Fig. 4b).

To evaluate the impact of FASN on viral replication, we generated two stable FASN knock-down (KD) A549 cell lines using different shRNA constructs. Western blot showed ~70% reduction in FASN protein expression level upon KD (73% for construct 1 and 66% for construct 2, Fig. 4c). Infection of these KD cells with wild-type WSN virus indicated a 10- to 25-fold drop in viral replication (Fig. 4c). Furthermore, we inhibited FASN function using three inhibitors, including C75, Fasnall and GSK2194069. Virus replication decreased significantly with an increasing concentration of all tested FASN inhibitor, while cell viability was unaffected under these concentrations (Fig. 4d; Supplementary Fig. 10).

To determine if the expression of NS1 protein can impact fatty acid synthesis directly, we used gas chromatography–mass spectrometry (GC-MS) to quantitatively measure de novo synthesized fatty acids using $^{13}C$ isotopomer enrichment analysis. We constructed inducible lentiviral vectors expressing influenza A virus NS1 (A/WSN/33 and A/Cal/04/09), PB2 (A/WSN/33), PA (A/WSN/33), NP (A/WSN/33), and GFP as a control. Lentiviruses were used to transduce A549 cells, which were induced to express the respective protein products with doxycycline for 24 h. Cells were then switched to complete media containing U13C-labeled glucose:unlabeled glucose = 1:1 (total glucose 2 g/L) for 24 h to label de novo synthesized lipids. Absolute amounts of de novo synthesized fatty acids, including myristic acid (14:0), palmitic acid (16:0), stearic acid (18:0), and cholesterol were determined by GC-MS. Iotopologue

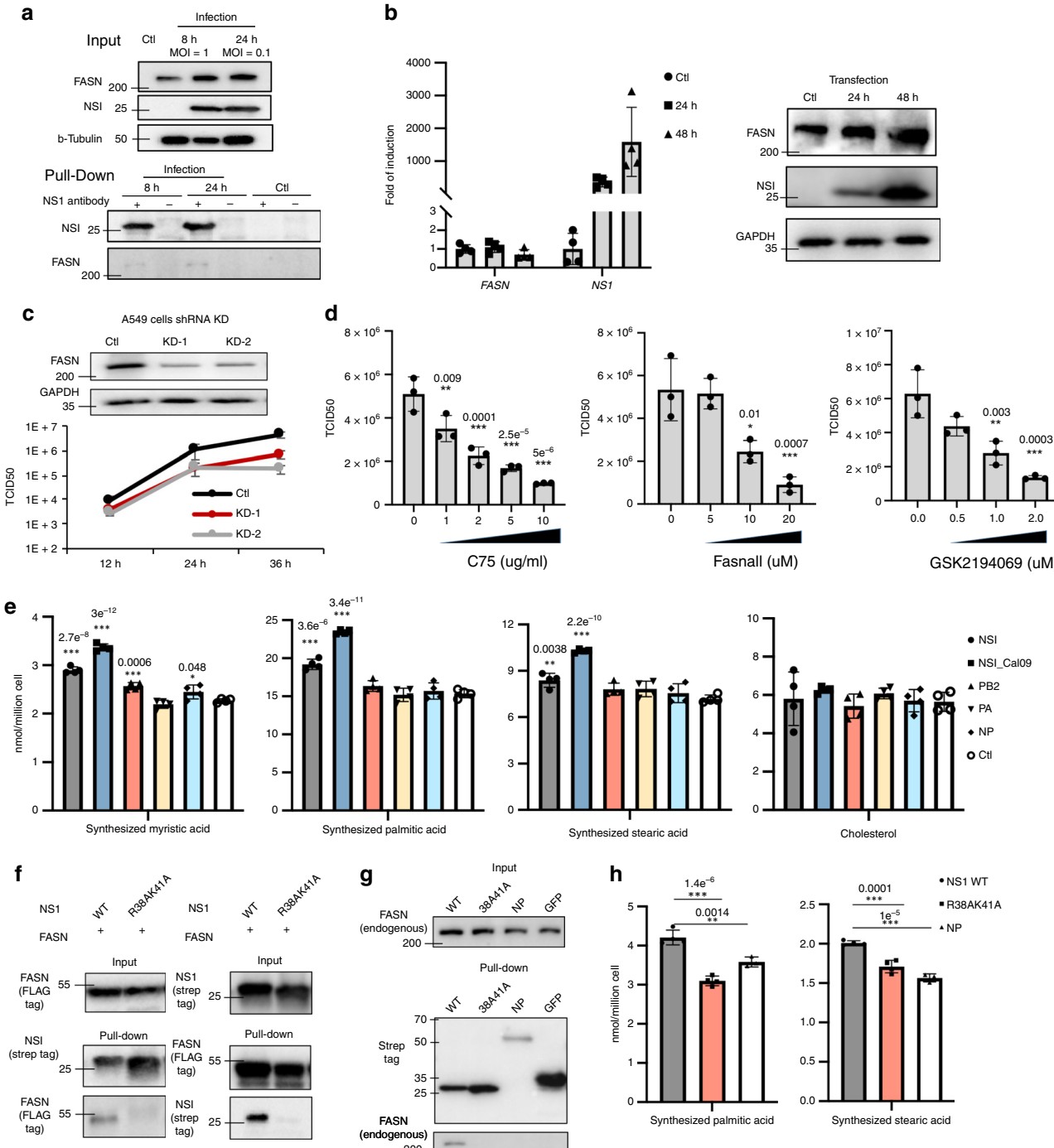

**Fig. 4 FASN is required for viral replication and regulated by NS1. a** Interactions between NS1 protein and FASN were examined by endogenous immunoprecipitation (IP)-western. Three biological replicates were performed, and a representative experiment is shown. **b** The gene expression level and protein expression level of FASN was examined post-NS1 overexpression in 293T cells ($N = 4$ biologically independent samples). **c** The effect of FASN on viral replication was examined using shRNA knock-down (KD) in A549 cells. The expression level of FASN upon KD was examined by western blot (upper panel). Control cells or KD cells were infected with WT WSN at MOI 0.1. The viral titer in supernatant were determined using TCID50 assays ($N = 3$ biologically independent samples). **d** The effect of FASN on viral replication in A549 cells was examined using FASN inhibitors, including C75, Fasnall and GSK2195069. Indicated concentrations of inhibitors were used at the time of infection. The effect of inhibitor treatment on viral replication at 24-h post infection was examined by TCID50 assay ($N = 3$ biologically independent samples). **e** The levels of newly synthesized fatty acids or cholesterol upon expression of indicated viral proteins were examined by GC/MS. Ctl is transfection with GFP-expressing vector ($N = 4$ biologically independent samples). **f** Interactions between WT and mutant (R38A/K41A) NS1 protein with FASN were examined by Co-immunoprecipitation assay, using Strep-tagged NS1 and FLAG-tagged FASN in 293T cells. **g** Interactions between NS1 protein and endogenous FASN were examined in A549 cells, which were transduced to be over-expressing indicated proteins. NP and GFP were used as control. Three biological replicates were performed, and a representative experiment is shown. **h** The levels of newly synthesized fatty acids upon WT or mutant (R38A/K41A) NS1 protein expression were examined by GC/MS ($N = 4$ for NS1 WT and mutant, $N = 3$ for NP, biologically independent samples). Data are presented as mean values $+/-$ SD for all panels. *$P < 0.05$, **$P < 0.01$, ***$P < 0.001$ (one-way analysis of variance (ANOVA) with Bonferroni multiple-comparisons test for panels **d** and **e**, two-tailed $t$ test for panel **h**, the exact $P$-value is labeled on the figure). Source data for all panels is provided in Source Data file.

distributions were fit using isotopomer spectral analysis (ISA)[48]. The NS1 proteins from both stains increased the de novo synthesis of all tested fatty acids relative to the GFP control (Fig. 4e; Supplementary Fig. 13). While a slight increase in myristic acid was observed upon PB2 expression, no other viral protein tested caused a significant change in any of the monitored fatty acids. Furthermore, we found that NS1 expression did not cause any significant change on cholesterol synthesis upon NS1 overexpression, whose levels is not controlled by FASN (Fig. 4e). These results suggest that the binding of NS1 to FASN results in the up-regulation of fatty acids synthesis and is beneficial to virus replication[46].

We next designed a series of HA-tagged NS1 truncation mutants to map the binding domain with FASN. Each truncation mutant was expressed alongside FLAG-tagged FASN in 293T cells, and immunoprecipitated with anti-HA affinity gel. The N-terminal region consisting of amino acids 1–73 was found to be necessary for the interaction (Supplementary Fig. 11a). Two residues in this domain, R38 and K41, are known to be important for RNA binding, as well as multiple protein binding, such as TRIM25, DDX21, and RIG-I[6,34,49,50]. Mutation of these residues led to a strong reduction in the interaction with FASN, assayed by Co-IP in 293T cells (Fig. 4f) or by precipitating endogenous FASN with A549 stable cell lines over-expressing different viral proteins (Fig. 4g). Functionally, the R38A/K41A mutation failed to upregulate FASN protein expression (Supplementary Fig. 11b), and resulted in a loss of NS1 function in upregulating de novo synthesize of palmitic acid and stearic acid (Fig. 4h; Supplementary Fig. 12). Notably, influenza NP protein is known to bind RNA, although we cannot rule out the potential mild interaction between NP and FASN due the low expression/pull down of NP (Fig. 4g), it did not modulate the function of FASN (Fig. 4e, g, h). Collectively, these data suggest that binding between NS1 and FASN is required for the IAV regulation of FASN function.

**Revealing the mechanism underlying an IFN-sensitive NS1 mutation.** As the enrichment of the exon library by md-LED gives a semi-quantitative evaluation of the binding efficiency, one potential application is the direct comparison of PPIs by different protein variants. Toward this end, we chose to investigate the interaction profile of a previously described NS1 mutant, D92Y, that was discovered in our high-throughput genetic screen for IFN-sensitive variants[51]. Cells infected with IAV containing the D92Y mutation produce higher amounts of IFN compared with infection with WT, indicating an inability of this mutant NS1 to inhibit IFN induction, but the mechanism of this loss-of-function is unknown. To determine if this phenotype could be due to a change in the PPI profile, we applied md-LED on the NS1 D92Y (strain A/WSN/33). Among the 25 NS1 binders identified to bind to the wild-type protein (Fig. 2a), we found that only one inter-actor, CPSF1, failed to bind the D92Y mutant (Fig. 5a). In comparison, the AP-MS-derived profiles of the wild-type protein and D92Y NS1 mutant revealed 80 proteins with loss of binding affinity to the mutant, including CPSF1 and the known CPSF complex member FIP1L1 (Supplementary Fig. 13a, Supplementary Data 2). Together, the data from md-LED and AP-MS suggest that the D92Y mutation results in a loss of binding to the CPSF complex. In support of this, immunoprecipitation of HA-tagged NS1 in 293T cells resulted in co-immunoprecipitation of Flag-tagged CPSF1, but not the D92Y mutant (Fig. 5b). Furthermore, purified CPSF1-FLAG was found to interact with NS1-HA in vitro, but this interaction was reduced by the D92Y mutation (Supplementary Fig. 13b).

CPSF1 is the largest component of CPSF complex, which is critical for pre-mRNA 3′ processing, cleavage, and poly(A) addition. By blocking the expression function of CPSF complex, NS1 inhibits cellular mRNA transport and protein expression, including the expression of antiviral interferon-stimulated genes[34,35]. To examine the inhibition of cellular protein expression, we used a GFP expression plasmid, pEGFP-C1, as a reporter. While wild-type NS1 efficiently down-regulated the expression of the GFP reporter, the D92Y mutant had no impact on GFP expression relative to the control (Fig. 5c). The reduced protein expression is consistent with reduced mRNA levels, as quantified by real-time PCR after reverse transcription with oligo-dT, relative to the expression of GAPDH (Fig. 5d). Over-expression of CPSF1 results in significant inhibition of wild-type influenza A virus replication, but not of the D92Y mutant virus, which already lacks CPSF complex recruitment (Fig. 5e).

CPSF1 is a large, multidomain protein and its binding interface with NS1 has not been previously mapped. To examine the binding sites, we evaluated the secondary structure and exon arrangements of CPSF1 and fragmented the protein into six regions that should still fold properly (Fig. 5f)[52–55]. All fragments were expressed well in 293T cells upon transient transfection. Immunoprecipitation of each fragment revealed that only fragment 1, corresponding to amino acids 1–313 and exons 1–8, pulled down NS1 (Fig. 5f). In the md-LED data, we observed that N-terminus of the protein indeed showed high-enrichment score, especially with exons 5 and 6 (Supplementary Fig. 14). Together, these results identify a interacting interface between NS1 and CPSF1 required for innate immune suppression.

## Discussion

Currently, AP-MS is one of the most commonly used and well-established methods for detecting protein–protein interactions. Although the sensitivity and accuracy of AP-MS continues to increase, some limitations remain. First, high-quality antibodies are required for efficient pull-down of the bait protein. This limitation can be partially circumvented by tagging the target protein with high-affinity epitopes, but it is often difficult to express tagged proteins in the cell type of interest, and may result in nonphysiological levels of expression. Second, as protein samples cannot be amplified, low abundance proteins may not reach the detection limit for mass spectrometry[56]. Third, the current technical reproducibility of mass spectrometry is generally not as high as next-generation sequencing. This is mainly associated with the high complexity of sample preparation, chromatography, and mass spectrometry procedures[57]. All AP-MS methods rely on complex scoring algorithms to determine the probability that an interaction is "real", each of which oftentimes gives very different answers. Finally, the complexity of the cellular proteome and limitations in detectable peptides makes comprehensive measurement of an input sample impossible. As such, relative quantification of AP-MS results relies on the inclusion of internal standards between samples.

In contrast, md-LED overcomes some of the above limitations by converting the task of detecting a large number of different proteins to that of detecting nuclear acid sequences. The relatively even distribution of the exon library enables equal representation of cellular proteins, regardless of their expression levels in specific types of cells. Moreover, the high-throughput, comprehensive sequencing of both the input and bound output allows for semi-quantitative evaluation of binding affinity and increases the confidence level of hit identification. Due to the isolation of specific protein domains, interactions that are identified are likely to be direct, simplifying downstream analysis and hypothesis generation.

md-LED is also complementary to currently used mRNA display methodologies using natural cDNA libraries[32,33]. Previously reported cDNA screening usually requires multiple rounds of

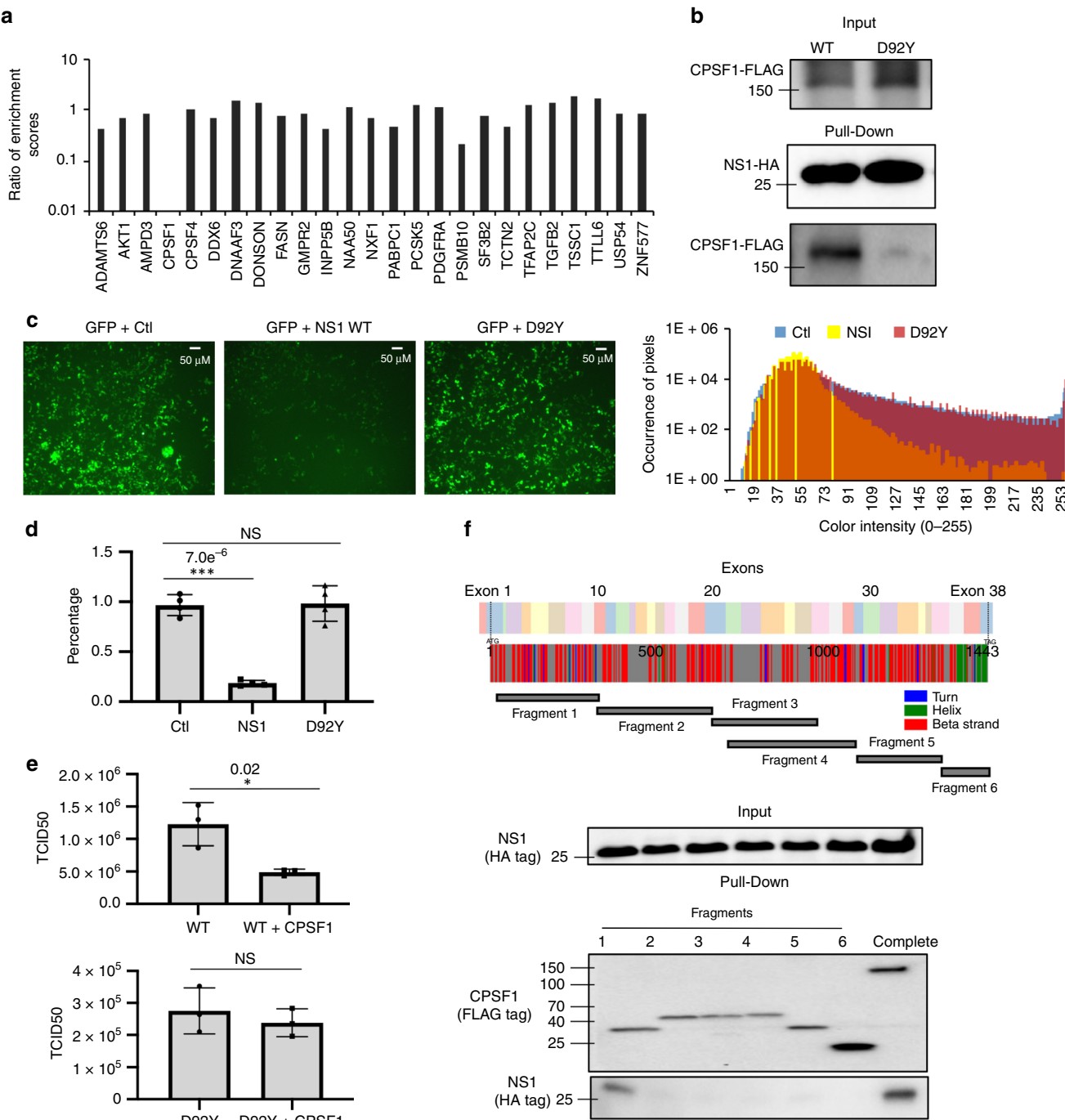

**Fig. 5 Mechanistic characterization of an NS1 IFN-sensitive mutation D92Y. a** Ratio of average enrichment scores of indicated proteins binding to NS1 versus D92Y mutant ($N = 6$ for NS1 protein and $N = 2$ for D92Y mutant, biologically independent samples). A lower ratio represented a stronger loss of binding for D92Y mutant. **b** Interactions between CPSF1 and NS1 proteins (WT and D92Y mutant) were examined by immunoprecipitation. **c** Inhibition of protein expression by WT or D92Y NS1 proteins using GFP reporter. GFP reporter was transfected in 293T cells together with WT or D92Y NS1 expression plasmid. Fluorescence intensity was examined 24-h post transfection. Empty vector was used as a control. Three biological replicates were performed, and a representative experiment is shown. The histogram of fluorescence intensity is shown on the right panel. K–S test of green channel intensity distribution between NS1 and Ctl shows $P < 2.2e^{-16}$. **d** Mature *GFP* mRNA was examined by poly-A-specific reverse transcription and real-time PCR relative to GAPDH. Empty vector was used as a control ($N = 4$ biologically independent samples). **e** Effect of CPSF1 overexpression on the replication of WT or D92Y mutant viruses ($N = 3$ biologically independent samples). Viral titer was examined by TCID50 assay. **f** Interactions between NS1 proteins and CPSF1 fragments were examined by immunoprecipitation (IP)-western. The arrangement of exons, secondary structures, and the range of amino acids of each fragment is shown. Three biological replicates were performed, and a representative experiment is shown. Data are presented as mean values $+/-$ SD for panels **d** and **e**. *$P < 0.05$, **$P < 0.01$, ***$P < 0.001$ (two-tailed $t$ test, the exact $P$-value is labeled on the figure). Source data for **a**, **b**, **d**, **e**, **f** is provided in Source Data file.

selection with specific elution conditions to decrease backgrounds. Genome-wide screening of a cDNA library using a general elution condition has not been reported, which indicates the challenge. Moreover, cDNA libraries are specific to the expression profile of the cells and need to be regenerated for each cell type and condition, although more tedious than the exonic library, provide additional information on tissue specificity for downstream phenotypic analysis. Combined applications leveraging both cDNA and md-LED input libraries may be useful in the future to extract complementary data sets.

While promising, several improvements could yet be made to the md-LED methodology to optimize it further. First and importantly, the current input library was constructed through exon enrichment of a randomly fragmented DNA library. As such, the translated protein fragments might contain flanking regions of adjacent introns. These flanking regions may affect the tertiary structures of real exons, and making the selected hits biased toward interactions that only require short peptides, or to highly stable protein motifs/domains whose structures are not easily disrupted by flanking sequences. To address this, we incorporated C-terminal FLAG tags for pre-enrichment and include internal comparisons to a GFP bait. We also calculated for the frequency of each exon regardless of the flanking sequence. One way to further improve this would be to utilize high-throughput DNA synthesis technology to accurately control the length and composition of input library. Second, md-LED is only able to characterize direct physical protein–protein interactions, and cannot detect interactions dependent on posttranslational modifications or tertiary folds that may cross multiple exons. Finally, similar to AP-MS and other affinity-based technologies, protein interactions with low affinity or of a transient nature will be difficult to detect. Genome-wide Bi-FC[58,59], proximity labeling, or protein correlation profiling methods[60] provide complementary approaches to examine transient interactions and might be able to address this issue.

In testing the md-LED method with the IAV protein NS1, we were unable to confirm the binding of NS1 with nine examined cellular proteins through Co-IP western, suggesting that binding of the individual domains may be restricted in the context of the full protein and the existence of false positives in our data sets. We were also unable to detect PIK3R2 through the md-LED method, which has been shown previously to be interact with the SH3 motif of NS1[61]. This may be due to an interaction interface that is dependent on multiple exonic sequences one of several other technical limitations discussed above. Thus, this new technology should be seen as highly complementary to existing approaches, and should be considered as one of several technologies available for use in the comprehensive identification of protein–protein interactions.

Viral infection often leads to changes in cellular lipid metabolism[62,63], and the essential role of fatty acids in viral replication is increasingly being appreciated[47,62–65]. Some viral proteins, including NS3 of dengue virus and NS4B of HCV, were shown to interact and modulate FASN activity[63–65]. Although previous reports have documented the regulation of lipid synthesis by influenza infection, we identifies NS1 as the viral protein that mediates this effect through the cellular partner FASN. Notably, cholesterol synthesis is not affected by NS1 and is not controlled by FASN, but is otherwise linked to the up-regulation of the IFN response and inhibition of viral infection[62].

Differential screening is an important approach to elucidate the mechanisms underlying loss-of-function mutations on the viruses and to uncover cellular functions that are activated or inhibited by viral proteins. md-LED allows us to explore the PPIs across such genetic mutants in a high-throughput manner. Using md-LED, we found that the IFN-inducing D92Y mutation results in a loss of NS1

interaction with CPSF1. These results were confirmed by AP-MS. Functionally, the D92Y mutant fails to recruit the CPSF complex and therefore fails to block host mRNA synthesis and induction of the IFN responsive genes. We expect that md-LED can be a new tool to interpret the functional variations among natural variants by quantitatively profiling their interactomes.

In conclusion, md-LED is a new method for the identification of PPIs. Powered by the sensitivity and unprecedented scale of high-throughput sequencing, md-LED could be leveraged to build cellular protein–protein interaction networks in massively parallel experiments. The quantitative nature of the data provides additional power for mechanistic interrogation in a manner highly complementary to traditional approaches.

## Methods

**Cells, viruses, and plasmids**. An eight-plasmid reverse genetics system of influenza A/WSN/33 virus (WSN) was utilized to reconstitute WT and mutant viruses. 293T cells were cultured in DMEM (Corning) with 10% FBS (Corning). A549 cells were cultured in RPMI 1640 (Corning) with 10% FBS (Corning). 293T cells were used for transfection of mammalian expression plasmid to overexpress viral and cellular proteins. A549 cells were used for transduction of lenti-virus vector expressing each bait protein for AP-MS and for immunoprecipitation of endogenous FASN. NS1 (WT and R38A/K41A mutation), PB2, PB1, PA, NP (from WSN strain), NS1 from Cal09 and GFP protein were cloned into Lenti-X Tet-one inducible expression plasmid with 2×Strep affinity tag at C-terminus. Cellular proteins with 1xFLAG tag were purchased from Harvard plasmid database, Origene, or amplified from cellular mRNA/cDNA and cloned into pCMV mammalian expression vector. NS1 protein with R38A/K41A or D92Y mutation was generated using a PCR-based site-directed mutagenesis strategy.

**Generation of exon library for mRNA display**. PBMCs were obtained from UCLA CARF Virology Core, collected from antonymous donors. Genomic DNAs were extracted using the DNeasy Blood and Tissue Kit (Qiagen). DNAs were fragmented using Covaris focused-ultrasonic technology, and size-selected with a range from 300–700 bp. The fragmented DNAs were end-repaired (NEB), dA-tailed using klenow exo- (NEB) and ligated with customized Y-shape adaptor as below. 5′-GGAGCCGCTACCCTTATCGTCGTCATCCTTGTAATCTGCCTGG CTTCCAGTGGAGCT; 3′-CCCTGTTAATGATAAATGTTAATGGTGGTACCG AAGGTCACCTCG-p.

Fragments were amplified and hybridized twice to human exon array using Roche NimbleGen SeqCapkit (Roche). The hybridized fragments were finally enriched with the customized primer linking to the Y-shape adaptor as below. A T7 promoter and Kozac sequence (5′ end), as well as a constant 3′ linker sequence encoding a Flag-tag were encoded in the adaptor sequence, for affinity purification from the in vitro translation reaction. T7Koz: TTCTAATACGACTCACTATAGG GACAATTACTATTTACAATTACCACCATGG Lib Rev: GGAGCCGCTACCC TTATCGTCG.

**Generation of cDNA library for mRNA display**. mRNAs from A549 cells were extracted using Trizol (ThermoFisher) and fragmented with 10 mM magnesium at 94 °C for 4 min. Reverse transcription was performed with super-script III system (Invitrogen) using poly-dT (IDT) as primers. Second strand synthesis was performed using mRNA second strand synthesis module (NEB). The resulting double strand DNAs were end-repaired, dA-tailed, and ligated with customized Y-shape adaptor, same as for the exon library.

**Expression and purification of bait proteins**. Open reading frames of viral proteins were cloned into pcDNA5 mammalian expression plasmid with a C-terminal HA tag. For each replicate, ~150 million 293T cells were transfected with 100 μg of DNA plasmid with calcium phosphate transfection reagents (Clotech). Cells were lysed at 2 days post transfection with binding buffer (50 mM Tris-HCl pH 7.4, 0.5% NP-40, 150 mM KCl, 1 mM EDTA, and protease inhibitors). Cell lysates were incubated with HA beads (Sigma-Aldrich) for overnight at 4 °C with constant agitation, and washed with wash buffer (50 mM Tris-HCl pH 7.4, 2% NP-40, 300 mM KCl, 1 mM EDTA, and protease inhibitors) for five times.

**mRNA display and sequencing**. For each reaction, the exon library (DNA templates) was transcribed by T7 run-off transcription (Ambion), and 1 nmole of mRNA was ligated to the pF30P linker (5′ phospho-A(21)-spacer phosphoramidite 9 (3)-ACC-puromycin, 1.2 nmoles) via the splint oligonucleotide (1.1 nmoles) by T4 DNA ligase (NEB) in a 200 μl of reaction. After purification and isolation of ligated mRNA templates, in vitro translation was performed using reticulocyte lysate (Ambion) in 100 μl of reaction volume followed by incubation with KCl (500 mM final) and MgCl$_2$ (60 mM final) for 30 min at room temperature to enhance fusion formation. The mRNA–protein fusions were then affinity-purified using M2 anti-Flag beads (Sigma-Aldrich) to remove sequences containing

nonsense mutations and non-fused RNA templates and proteins. After elution with 3 × Flag peptides, the fusions were reverse transcribed with superscript III (Invitrogen), and a fraction of the purified sample was reserved to determine the frequencies of each coding sequence in the input library. The purified fusion sample was incubated with bait protein for 3 h at 4 °C. After washing, the immobilized fusion samples were eluted by heat (95 °C) and PCR amplified using the following primers (T7-Rec, Lib Rev). The amplified DNA fragments from input and post selection were then prepared for high-throughput sequencing using lllumina Hiseq PE150. Barcodes of 6 bps were added to distinguish among different samples. Six biological replicates were performed for each bait protein, including in vitro transcription, translation, and enrichment steps. T7-Rec: GGGACAATTACTATT TACAATTACCACCATGG Lib Rev: GGAGCCGCTACCCTTATCGTCG.

**Sequencing data analysis**. Data were analyzed by customized bash and python scripts. Paired-end fastq reads were de-multiplexed into corresponding samples by the 6-bp barcodes. Reads were mapped to human reference genome (hg19) using Tophat2 using default parameters. Unmatched or multi-matched (>2) reads were filtered out. Coding frame of each fragment was analyzed by the mapping position and corresponding CDS position. Only the reads with correct orientation and frame for defined ORFs on cDNAs were considered. The reads counts for each CDS and transcript features were calculated using bamtools. The enrichment score for each CDS was calculated as the relative frequency of a particular CDS in the selection pool to that in the input library. We also evaluated the enrichment of transcripts by summarizing the frequency of all related CDSs in the selection library, to that in the input library.

$$\text{Enrichment score}_{CDS}$$
$$= \text{relative frequency in selection library}_{CDS} / \text{relative frequency in input library}_{CDS} \tag{1}$$

$$\text{Enrichment score}_{transcript}$$
$$= \sum \text{relative frequency in selection library}_{CDS} / \sum \text{relative frequency in input library}_{CDS}. \tag{2}$$

To reduce he noise, only the genes showed to have both enriched CDSs (the highest enrichment score of CDS of corresponding gene >2) and enriched transcripts (enrichment score >1), and were consistent among replicates were considered as hits for the corresponding bait protein. Genes that showed enrichment in GFP bait control were considered to be noise and filtered out. The enrichment score of a specific gene was calculated as the highest enrichment score of the corresponding functional transcripts, representing the binding possibility of corresponding cellular gene to the bait protein.

$$\text{Enrichment score}_{gene} = \max(\text{enrichment score}_{transcripts}) \tag{3}$$

**Affinity purification–mass spectrometry (AP-MS)**. WT and D92Y mutant NS1 ORFs were cloned into the pLVX-TetOne-Puro vector with 2×Strep tag at the N-terminus. A 2×Strep-tagged GFP was cloned as a control. Lentiviruses were generated by transfecting 293T cells with pLVX-TetOne-Puro vector, Gag-Pol packaging construct, and VSV-G envelope. A549 cells were transduced with the generated lentiviruses and selected under 1 µg/ml puromycin. The expression of desired viral proteins was confirmed by western blot.

For immunoprecipitation, A549 cells expressing different viral proteins (or the control GFP protein) were induced with 1 µg/ml Dox for 24 h. They were left untreated or treated with 1000 U/ml type I interferon for 12 h before harvesting. Cells were lysed, cleared of cellular debris, and bound with 20 µL of Strep-Tactin Sepharose beads (IBA Lifesciences) in 550 µL IP buffer (50 mM Tris-HCl, pH 7.4, 150 mM NaCl, and 1 mM EDTA). Beads were washed four times (two times with 0.05% NP-40, and two times without) prior to on-bead protein digest. Streptactin-purified proteins were reduced and alkylated on beads with 20 µL reduction-alkylation buffer [50 mM Tris-HCl, pH 8.0, 2 M urea, 1 mM dithiothreitol (DTT), 3 mM iodoacetamide]. An additional 3 mM DTT was added to quench the reaction, and proteins were digested with 0.75 µg trypsin (Invitrogen). Formic acid was added to a final concentration of 1% to acidify the peptides. Peptides were desalted using Agilent OMIX C18 10 µL tips.

Digested peptides were subjected to LC-MS/MS analysis using an Easy-nLC 1000 coupled to a dual-pressure linear ion trap (Velos Pro) Orbitrap Elite mass spectrometer (ThermoFisher Scientific, San Jose, CA). Peptides were eluted by a gradient of 5–30% acetonitrile in 0.1% formic acid over 110 min delivered at a flow rate of 300 nL/minute. For each cycle, one full MS scan (150–1500 m/z, resolution of 120,000) in the Orbitrap was followed by 20 data-dependent MS/MS scans fragmented by normalized collision energy (setting of 35%) and acquired in the linear ion trap. Raw MS files were analyzed by MaxQuant version 1.3.0.3 and MS/MS spectra searched by the Andromeda search engine against a database containing reviewed SwissProt human and influenza protein sequences (20,226 total). MiST scoring algorithm was used to assign scores to bait–prey interactions against the GFP controls.

**Network and GO analysis**. Gene ontology enrichment analysis was performed through metascape. STRING was used as cellular PPI databases[39]. Network analysis was performed using networkx package in python.

**Gas chromatography–mass spectrometry (GC-MS)**. Cells were cultured in a 1:1 ratio of U$^{13}$C glucose tracer for 24 h. Prior to collection, cells were imaged on Molecular Devices ImageXpress XL to assess cell numbers. Then cells were dissolved in 6 M guanidine HCl and transferred to glass tubes for derivitiziation with 3 M methanolic guanidine HCl. Samples were prepared alongside standard curve samples made up of FAMES mix (Nu-chek Prep, GLC 20a) and Cholesterol (Sigma, C8667).

Total cellular fatty acids were prepared by mild acid methanolysis[66] with the modifications. Integration and quantification was performed on MassHunter Quantitative Analysis Program (Agilent Technologies, B.06.00). Analysis for total quantification of fatty acids and cholesterol and relative contributions of synthesis to the respective pool over labeling period were determined by fitting the isotopologue distributions to isotopomer spectral analysis (ISA).

**Examining the cellular abundance of detected binders**. Protein abundance of binders detected by md-LED method and AP-MS were compared. Mass spectrometry raw files of A549 cellular proteome were downloaded from PRIDE (project: PXD000418). Raw files were searched against Unipro human proteome database using ProLuCID search engine with protein FDR < 0.01. "AP-MS" protein category was identified in our NS1-AP-MS experiments with score >0.8 in either IFN treated or nontreated A549 cells. A549 RNA-seq quantification with three biological replicates was downloaded from ENCODE (experiment ENCSR937WIG).

**Immunoprecipitation and in vitro binding**. Immunoprecipitation experiments were performed with HA- and FLAG-tagged proteins expressed in 293T cells. Briefly, cells were transfected with corresponding expression plasmids with lipofectamine 2000 reagents (Invitrogen), and lysed at 2 days post transfection with RIPA buffer (50 mM Tris-HCl pH 7.4, 0.5% NP-40, 150 mM NaCl, 1 mM EDTA and protease inhibitors). For binding experiments with RNase A treatment, 0.1 mg/ml RNase A (ThermoFisher Scientific) were added into cell lysis buffer for 30 min before immunoprecipitation. Cell lysates were incubated with 1 µg anti-FLAG for overnight at 4 °C with constant agitation, washed with RIPA buffer five times, and eluted with 60 µl of SDS-PAGE sample buffer. All samples were subjected to SDS-PAGE and western blotting analysis.

For in vitro binding experiments, FLAG-tagged CPSF1 was expressed in 293T cells. Cells were lysed at 2 days post transfection with RIPA buffer, and bound with anti-FLAG antibody for 4 h at 4 °C with constant agitation. Then CPSF1 protein was eluted with 3× FLAG peptide overnight at 4 °C with constant agitation. HA-tagged WT or D92Y mutant NS1 protein was expressed in 293T cells, lysed at 2 days post transfection and bound with anti-HA overnight at 4 °C. They were washed five times with RIPA buffer, and incubated with eluted CPSF1 protein for 4 h at 4 °C. Samples were then washed with RIPA buffer five times, and eluted with 60 µl of SDS-PAGE sample buffer. All samples were subjected to SDS-PAGE and western blotting analysis.

**Western blotting**. Proteins in SDS-PAGE sample buffer were heated at 95 °C, resolved by SDS-PAGE gel electrophoresis, and then transferred onto the PVDF membrane. The following antibodies were used: DYKDDDDK tag monoclonal antibody (FG4R, ThermoFisher, Catalog # MA1-91878; 1:5000), monoclonal anti-Strep tag antibody (Sigma-Aldrich, Cat.SAB2702216-100UL; 1:5000), HA tag monoclonal antibody (2-2.2.14, ThermoFisher, Catalog no. 26183; 1:5000), polyclonal influenza A virus NS1 (nonstructural protein) antibody (GeneTex, cat no. GTX125990; 1:1000), monoclonal influenza A virus NS1 (nonstructural protein) antibody (GT1653, GeneTex, cat. no. GTX633685; 1:1000), fatty acid synthase (C20G5) Rabbit mAb (CST, cat. no. 3180 S; 1:1000), monoclonal anti-GAPDH antibody (Sigma, cat no.: G8795-200UL; 1:5000), and beta tubulin monoclonal antibody (ThermoFisher, catalog no. 32-2600; 1:5000).

**shRNA**. shRNA against FASN is constructed using the following sequences:
shFASN-Construct 1-F
ccggcGAGAGCACCTTTGATGACATCTCGAGATGTCATCAAAGGT GCTCTCGtttttg
shFASN-Construct 1-R
aattcaaaaaCGAGAGCACCTTTGATGACATCTCGAGATGTCATCAAAG GTGCTCTCG
shFASN-Construct 2-F
ccggcCCCAGGCTGAAGTTTACAACTCGAGTTGTAAACTTCAGCCT GGGtttttg
shFASN-Construct 2-R
aattcaaaaaCCCAGGCTGAAGTTTACAACTCGAGTTGTAAACTTCAGCCT GGG.
Non-mammalian shRNA control (SHC002) were used as scramble control. Stable knock-down A549 cell lines were constructed by infecting with lenti-virus and selection with 1ug/ml puromycin for 1 week. Western blot were used to determine knock-down efficiency.

**Viral replication assay (TCID50, viral copy number, mCherry reporter)**.
TCID50 assay were performed in A549 cells by observing cellular cytotoxic effect
(CPE). Viral copy numbers were measured using real-time PCR using standard
curve with the following primer targeting NP segment[67].

NP-forward: GAC GAT GCA ACG GCT GGT CTG

NP-reverse: ACC ATT GTT CCA ACT CCT TT.

In total, 50 ng virus-inducible mCherry reporter were transfected in
293T cells in 24-well plates[68]. Media were changed 24 h post transfection, and cells
were infected with indicated virus for 24 h. The mCherry signal was observed
under fluorescence microscope, as an indicator of the replication capacity of virus.

**Generation of mutant viruses**. Individual mutant viral plasmids were generated
by quick-change system. To generate the mutant virus, ~2 million 293T cells were
transfected with 10 μg DNA. To measure the growth curve, ~1 million A549 cells
were infected with MOI 0.1, and supernatants were collected at the indicated time
to measure viral titer.

**Quantification of mRNA (GFP, poly-A)**. 293T cells were transfected with WT or
mutant NS1 protein (100 ng) and GFP reporter (10 ng). At 24 h post transfection,
total cellular RNAs were extracted from infected cells with the Purelink RNA Mini
Kit (Ambion), and reverse transcribed by Superscript III Reverse Transcriptase
(ThermoFisher) using oligo-dT as primer. Quantitative real-time PCR was per-
formed using Taq polymerase and SYBG.

**Fragmentation of CPSF1**. To avoid disrupting protein structures and restricting
protein expression, we evaluated the secondary structure and intrinsic unstructured
region of CPSF1, and fragmented the entire protein into 6 small regions. Six
fragments of CPSF1 were constructed by cloning the corresponding regions into
pCMV6 vector.

**Statistical analysis**. Detailed statistical analysis were included in each figure legend.
Results were analyzed by two-tailed Student's $t$ test. Differences were considered
statistically significant when $P < 0.05$ (*), $P < 0.01$ (**), or $P < 0.001$(***).

**Reporting summary**. Further information on research design is available in
the Nature Research Reporting Summary linked to this article.

## Data availability

Scored mass spectrometry data files have been uploaded as Supplementary Data 2. The
sequencing data are deposited to NIH Short Read Archive (SRA) with accession numbers
PRJNA520773 and PRJNA383938. The mass spectrometry proteomics data of current
study have been deposited to the ProteomeXchange Consortium via the PRIDE partner
repository with the data set identifier PXD018377. A complete list of all primers used in
this study is provided in Supplementary Table 1. Source data for Fig. 2a, Fig. 3b, c and d
are provided as Supplementary Data 1 and 3. Source data for Fig. 2b, Fig. 3d, Fig. 4a–h,
Fig. 5b, d, e, f are provided as a Source Data file. Databases used in this study also include:
PRIDE (project: PXD000418) and PDB: 2RHK.

## Code availability

Cutadapt (version 2.1) were used to trim off adaptor sequence, Bowtie2 was used for
mapping, bedtools (v2.17.0) were used to intersect mapped reads with exons. Custom
python scripts for further data analysis were deposited to github [https://github.com/
YushenDu/PED-].

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

## Acknowledgements

Funding support of NIH PO1 CA177322 to R.S., S.C., and N.K. and NIH U19AI135972.

## Author contributions

Y.D., J.F.H., A.O., S.B., N.J.K., and R.S. designed the experiments. Y.D., J.F., Q.Z., A.O., Y.T., T.H.Z., M.H., K.T., L.C., X.M., M.J.M., L.M.S., L.D., and D.G. performed the experiments. X.L. performed the high-throughput sequencing. Y.D., J.F., T.H.Z., S.C., and L.D. analyzed the data. Y.D., J.F.H., S.B., N.J.K., and R.S. drafted and/or revised the papers.

## Competing interests

The authors declare no competing interests.
