## [Peer Review File · Nature Communications]

Reviewers' Comments:

Reviewer #1:

Remarks to the Author:

Du et al. developed an Exon Display (PED) technology to detect cellular protein-protein interactions. To validate this proof of concept, the authors applied the PED to Influenza A Virus (IAV) to detect what cellular factors are associated with the non-structural protein 1 (NS1). The authors identified 25 high confidence hits, including several known interacting proteins, such as CPSFs. Eight of the ten interactions were confirmed by co-IP. Furthermore, the new NS1-interacting host factor, FASN was chosen for more in-depth study. They showed that NS1 directly regulate global lipid metabolism. In addition, they compared the cellular proteins interacting with wild type NS1 vs. the D92Y mutant. They found D92Y failed to interact with CPSF1. Although this study presents a new method that might be complementary for the classic AP-MS, there are several major concerns (see below).

Major concerns:

The authors claimed that the PED has the potential advantages of identifying low abundance interactors and in facilitating the identification of binding domains. In theory, it might be. But the data didn't support the authors' claim.

1. The authors confirmed 8 interactions by co-IP. But the reviewer doesn't think any of these 8 host factors has low expression level. If the authors want to support their point, they should choose low expression candidates.
2. Fig. 3E. The reviewer doesn't agree with the authors' conclusion. Take CPSF4 for example, exons 1, 2 & 8 all are critical for binding according to the presented data. These data not only fail to identify the specific binding domain but also challenge the proof of concept of the PED.
3. Fig. 4. is to show the functional relationship between NS1 and FASN. First, the authors must show the protein interaction between endogenous FASN and NS1 during viral infection. Secondly, the authors must show the co-localization between NS1 and endogenous FASN.
4. Fig. 4B. There is only 3-4 fold difference of TCID50 with vs. without drug, which is a minimal change to a virologist.
5. The RNAi experiment was done in 293T cells. The authors need to repeat the experiments in more physiological cells, such as A549 and primary lung epithelial cells.
6. The authors showed FASN was required for IAV replication and NS1 regulated global lipid metabolism. But there is no mechanistic study to show how NS1 regulates lipid metabolism by interacting with FASN.
7. It is not clear how the authors to determine the non-specific binding proteins during the IP step using protein G beads in the PED. It seems the authors only subtracted the GFP data.

Minor concerns:

1. The correspondence email should be an academic email address.
2. Line 577. "binding experiments with NP". It should be NS1.
3. Fig.2B. It was confusing at the first look because the authors didn't indicate the non-specific band, which might be IgG light chain.
4. Line 207. Are the 115 proteins ISGs as they only appeared after IFN treatment?

Reviewer #2:

Remarks to the Author:

I have studied only the lipidomics part of this manuscript (as requested by the editor) and I have only one comment:

Line 271: "...media containing 50% U13C-glucose..". This is misleading to non-expert readers. Please say e.g. "...containing U13C-glucose/unlabeled glucose 1:1 (?? mg/ml)".

Reviewer #3:

Remarks to the Author:

Du et al. established a novel protein-protein interaction screen termed Protein interaction detection by Exon Display (PED). The advantage of this method is the detection of low abundance binders (no bias towards unfavorable protein abundances in normal IP-based assays) and identification of specific domains. The authors evaluated the system by screening the interactome of the influenza virus NS1 protein. This revealed a large number of already described binding partners (but not all) and additional new ones. In functional studies the role of two factors were analyzed in more detail.

As expected from this group, this is a technical very sound study. The authors nicely describe and document the advantages and certain limitations of PED. The Enrichment scores in predicting the domain of a cellular protein with NS1 is not always obvious. The importance of the fatty acid synthetase (FASN) is not as obvious as the authors want.

Specific points:

- 1) Fig. 3E: Without knowing which domain of CPSF4 is interacting with NS1, the enrichment scores would not help very much. Is it possible to improve this?
- 2) Fig. 4b: The authors mention that FASN is required for viral growth, however, inhibition of viral replication is only 4-fold (despite favorable presentation of the results) and only tested in one cell line. It would be nice to substantiate this finding by using more appropriate cell lines (A549...).
- 3) Fig. 4F and G: Based on the enrichment scores it is unclear why fragment 3 (and to a certain degree fragment 5) is not binding to CPSF1. Thus, the enrichment score is a poor prediction of possible interacting domains (see point 1).

Reviewer #4:

Remarks to the Author:

See attachment

This manuscript by Du and coworkers reported the use of one round of mRNA display selection coupled with high throughput DNA sequencing to identify potential human intracellular protein domains that bind to the influenza A virus (IAS) NS1 or its mutant, from a genomic human DNA fragment library with a size of 300-700 bp. 25 proteins were enriched from multiple replicates, including four previously reported NS1-binders (CPSF4, CPSF1, PABPC1, and NFX1). They performed co-IP studies of 10 selected proteins and found 8 appeared to directly interact with NS1, including three previously unreported NS1-binders (PSMB10, AKT1, and FASN). Preliminary biochemical analyses were performed to examine the possible roles of FASN/NS1 and CPSF1/NS1(D92Y) in the viral host interactions. In general, the work is interesting, but the major claims were not supported by the data provided in the manuscript.

1. One of the major challenges in the generation of exon library is how to get nucleic acid sequences that do not contain intron regions, since unremoved introns could be used for coding by the protein synthesis machinery, introducing artificial peptide motifs that do not exist in natural proteome. Whereas this is not a problem by using natural cDNA libraries derived from matured mRNAs in which introns are absent, the use of genomic DNA as the starting material might have this issue. The 'human exon library' used in this work was generated by random fragmentation of genomic DNA and filtered to a size of 300-700 bp, followed by exon enrichment. The major concern is whether this is a real exon library or just a protein fragment library in which exons might contain a small fragment of peptide motifs coded by introns?

Using the exon 3 of CPSF4 that the authors found to be the highest enriched segments in CPSF4 (Fig. 3E) as an example:

According to the genomic structure of CPSF4 from accession# EF191081.1, the CDS of CPSF4 are joined by the following regions:

2161..2263,7867..7917,11199..11351,13354..13449,13780..13873,15446..15518,17044..17214,19510..19578. The exon 3 is coded by 153 bps (from 11199 to 11351). Unless there is an effective mechanism to distinguish and remove the intron sequences, the translation will continue at the end of the last codon of exon 3, resulting in translation of the intron (11352 to 13353) between exon 3 and exon 4 until a stop codon is encountered, resulting in the following amino acid sequence (exon 3 underlined):

GMCPFRHISGEKTVVCKHWLRGLCKKGDQCEFLHEYDMTKMPECYFYSKF**GKAPGALEALLRTRVQRGPLAAQCPH**
CLCLLFPSPYSQRRTLAEPASLSQLLGDVAIFCSSLFFFFFF*

This is very different from the natural exon 3 (underlined)-exon4 (italicized) fragment with the following amino acid sequence:

GMCPFRHISGEKTVVCKHWLRGLCKKGDQCEFLHEYDMTKMPECYFYSKF*GECSNKECPFLHIDPESKIKDCPWYDRG*
FCKH

The problem is that the bolded polypeptide sequence is not only artificial and absent in the natural CPSF4, but also its presence at the C-terminus of exon 3 could alter the properties of exon 3 polypeptide, including its binding affinity, specificity, and mode with other proteins. It's well known that the properties of polypeptides could be drastically changed due to the presence of several more residues at the N- or C-terminus. It is possible that the authors were lucky that the CPSF4 exon 3 sequences they isolated from the library still bound to NS1. Nevertheless, the binding properties of numerous other exon sequences could be altered due to the presence of the N- or C- terminal artificial motifs derived from introns, resulting in artifacts that are very hard to interpret.

With that said, it is unclear whether the hybridization exon enrichment step that the authors reported in the protocol allowed the complete removal of the exon sequences that contain part of the introns? If yes, detailed explanation is needed, and related data should be provided.

2. People in the field hypothesized that the use of less biased library with more evenly distributed input library could facilitate the identification of interactors with low abundance as suggested by previous studies, including those using mRNA display approach. However, a number of parameters could affect the identification of low abundance interactors, including the copy number of the interactor genes, the binding affinity, specificity, mode, and binding conditions. It is not clear whether the three novel NS1 binders identified by the authors can be classified as low abundance genes. More importantly, binding affinity and specificity data should be provided to further understand the putative interactions.

3. Compared to AP-MS, much less interactors were identified, despite the dataset could be extended based on the results from other methods. In this reviewer's opinion, the strength of this paper is not the method, but the novel NS1 binders that were not reported previously and the related biological significance. Only 10 selected proteins were tested for co-IP. Why not all 25 proteins so that more novel and hopefully physiologically significant interactors could be revealed? Pull-down assay and binding affinity analyses can be easily performed for that number of interactors.

4. Some advantages claimed by the authors are not unique to the method described in this manuscript. Tagging the target proteins is widely used in many proteomic searches. Current mass spectrometry allows for the identification of natural proteins with very low abundance. Lack of tissue type information is actually a disadvantage. Single round of selection coupled with HTS can also be applied to the mRNA display using natural cDNA library, and the biased problem could be addressed by different approaches.

Re: NCOMMS-19-08520

We would like to thank the reviewers for their great questions and suggestions for our study. We have been thorough in our attempts to address all the points raised by the reviewers. We have revised 8 main figure panels and added 8 supplementary figure panels. The main text is revised accordingly. We also included the abstract, main figures, figure legends and brief description of results on the identification of a cellular protein target of a Chinese medicine and mechanism. This separate study, provided for the preview of editors and reviewers, further supports our claims of this manuscript. Our point-by-point responses to the reviewers' concerns are provided in the following:

Reviewers' comments:

Reviewer #1 (Remarks to the Author):

Du et al. developed an Exon Display (PED) technology to detect cellular protein-protein interactions. To validate this proof of concept, the authors applied the PED to Influenza A Virus (IAV) to detect what cellular factors are associated with the non-structural protein 1 (NS1). The authors identified 25 high confidence hits, including several known interacting proteins, such as CPSFs. Eight of the ten interactions were confirmed by co-IP. Furthermore, the new NS1-interacting host factor, FASN was chosen for more in-depth study. They showed that NS1 directly regulate global lipid metabolism. In addition, they compared the cellular proteins interacting with wild type NS1 vs. the D92Y mutant. They found D92Y failed to interact with CPSF1. Although this study presents a new method that might be complementary for the classic AP-MS, there are several major concerns (see below).

Major concerns:

The authors claimed that the PED has the potential advantages of identifying low abundance interactors and in facilitating the identification of binding domains. In theory, it might be. But the data didn't support the authors' claim.

1. The authors confirmed 8 interactions by co-IP. But the reviewer doesn't think any of these 8 host factors has low expression level. If the authors want to support their point, they should choose low expression candidates.

Response: We thank the reviewer for the question. We agree with the reviewer that to better support our claim of PED's capacity of identifying low expression binders, we should

pick more candidates for validation. Previously we chose 10 genes which include previously identified binders, as well as novel binders that may have functional roles in pathways related to influenza virus replication, without consideration of their relative expression level. In the current revised version, we included all 25 hits, expressed the majority of the hits (23 out of 25) in 293T cells, and performed a Co-IP assay to examine the binding with NS1 protein.

As shown in Figure 2B, 14 binders with varying affinities were found to co-precipitate NS1. Based on the affinity and quality of pull down detected with western blot, we categorize them into strong (CPSF4 and binders with similar binding strength, including CPSF1, NXF1, PSMB10 and TFAP2C); medium (AKT1, DDX6, PABPC1, FASN, SF3B2, and TTL6), and weak binders (evidence of co-precipitation was not identified consistently in all four replicates (DNAAF3, INPP5B, and USP54) (Table S1). Notably, we identified and validated nine unreported NS1 interactors, including PSMB10, AKT1, FASN, DDX6, INPP5B, DNAAF3, TFAP2C, TTL6 and USP54 (Table S1). Among those, PSMB10, AKT1, FASN, DDX6, TFAP2C and TTL6 were considered strong-to-medium level binders.

We further compared the cellular abundance of identified binders in our matched PED and AP-MS datasets. Consistent with our previous observation, a significant lower abundance of identified PED interactors identified using PED was shown in both the whole cellular proteome dataset and RNA-seq data, when compared with AP-MS interactors identified with AP-MS (Figure 3D).

2. Fig. 3E. The reviewer doesn't agree with the authors' conclusion. Take CPSF4 for example, exons 1, 2 & 8 all are critical for binding according to the presented data. These data not only fail to identify the specific binding domain but also challenge the proof of concept of the PED.

Response: We thank the reviewer for this comment. The expression and enrichment of individual exon and exon fragments theoretically allows for the identification of the binding domain of the prey. As evidenced in Figures S9 and S15, data from this approach did lead to the enrichment of previously reported domains as well as new interfaces confirmed by deletion mapping. That being said, we agree with reviewer's concern that this approach may not be ideal for accurate de novo domain identification due to a number of interfering factors. Firstly, not all binding sites are contiguous, so interactions and interacting domains won't always be faithfully represented by the library. Secondly, as a single exon or exon fragment may not fold or behave in the same way as it is in the complete protein, not all interfaces can

be captured cleanly. Thirdly, while noise within the dataset can be partially counteracted by analysis of fragments across an entire gene, the level of noise for individual fragments is high enough to potentially confound identification. For example, the data in Figure S15 could indicate an enrichment in exon 23 and 33, which is not validated by the deletion mapping. Thus, as the reviewer suggested, we have turned down the point that we can accurately identify the binding domain using PED. We have also included a more nuanced discussion of the use of this approach for domain identification.

However, we still argue that PED can at least indicate the possible binding domains of some proteins. Besides the evidence shown in our current paper, we have another project using the same PED approach for the identification of drug targets. We included an abstract and five figures from this project at the end of this response letter. In a few words, we used PED to identify RPL27A as the cellular target of Martynoside (MAR). MAR attenuated 5-FU induced down-regulation of RPL27A through stabilizing RPL27A. From the PED exon enrichment data, we observed that the exons 4 and 5 of RPL27A may mediate the interaction. Using the exons 4 and 5 as the binding pocket, we performed docking simulation and proposed the potential binding mode between MAR and RPL27A. Quantitative mass spectrometry showed that the addition of MAR reduced the ubiquitination on amino acid K92 and K94 of RPL27A, consistent with the predicted drug-docking model. Based on the above results, we further defined the functional mechanism of MAR binding to RPL27A, as details attached.

Beyond the scope of this paper, we are also working on improving the input library from a random fragmented human exon library to a rational designed oligo library through parallel chemical synthesis. We anticipate that a more precisely controlled and less diverse input library will further enable the identification of binding domains.

3. Fig. 4. is to show the functional relationship between NS1 and FASN. First, the authors must show the protein interaction between endogenous FASN and NS1 during viral infection. Secondly, the authors must show the co-localization between NS1 and endogenous FASN.

Response: We thank the reviewer for the suggestion. We performed the endogenous IP by infecting A549 cells with influenza virus vs a mock infection. Endogenous FASN was pull-down with Protein G beads conjugated with anti-NS1 antibody. FASN was indeed detected by NS1 pull-down, but not with the non-infected cells or IgG controls (Figure 4A). However, the co-localization is tricky as both of the proteins are multi-functional and have

multiple protein binders. 167 physical interactors with FASN were reported in BioGRID (<https://thebiogrid.org/108488/summary/homo-sapiens/fasn.html>). NS1 also possesses multiple cellular interactions, depending on the stage of viral infection¹⁻³. Thus, despite the use of high-resolution microscopy (SIM), we were not able to observe a statistically significant co-localization between FASN and NS1 in infected A549 cells.

To show the functional importance of the physical interaction, in addition to the endogenous IP, we identified the potential binding domain through truncation mapping and specific mutations that result in loss of binding. We showed that the N terminal domain of NS1 is the major contributor to FASN binding, and that a double mutation on this domain: R38A/K41A almost eliminated binding. Functionally, R38A/K41A showed reduced ability to induce FASN up-regulation. It also had reduced capacity to induce fatty acid synthesis as quantified by GC-MS.

The above data suggest that NS1 and FASN interact during viral infection, and this physical interaction is functionally important. By disrupting functional interaction through a mutation, NS1's ability to upregulate FASN is diminished.

4. Fig. 4B. There is only 3-4 fold difference of TCID50 with vs. without drug, which is a minimal change to a virologist.

Response: We agree that for virologists, inhibition of viral replication in the log scale is often expected, especially with anti-viral drugs. Here we are targeting a regulatory cellular protein. Previously we only used C75 as an inhibitor to examine the importance of FASN. However, C75 has high cell toxicity, making it difficult to use at a high concentration. To further evaluate the requirement of FASN for influenza viral replication, we established two stable FASN knock-down (KD) A549 cell lines using different shRNA constructs. The western blot showed ~70% reduction in FASN protein expression level upon KD (Figure 4C). Infection of these cells with wild type influenza virus (strain A/WSN/33) indicated a drop in viral replication with decreased FASN protein expression (Figure 4C). At 36h post infection, we detected a 10-25-fold drop in viral titer as quantified by TCID50 assay. Furthermore, we inhibited FASN function using three different inhibitors: C75, Fasnall and GSK2194069. Virus replication decreased significantly with an increasing concentration of all tested FASN inhibitors, while cell viability was unaffected under these concentrations (Figure 4D, Figure S11).

5. The RNAi experiment was done in 293T cells. The authors need to repeat the experiments in more physiological cells, such as A549 and primary lung epithelial cells.

Response: Thanks, and we agree with the reviewer on this point. As stated above, we have now established stable FASN knock-down A549 cell lines using two different shRNA constructs (Figure 4C). In KD cells, we detected a near 10-fold reduction of viral titer at 48h post infection.

6. The authors showed FASN was required for IAV replication and NS1 regulated global lipid metabolism. But there is no mechanistic study to show how NS1 regulates lipid metabolism by interacting with FASN.

Response: We thank the reviewer for the question. In the previous version, we observed that FASN protein expression level was increased upon NS1 over-expression (Figure 4B). The RNA expression level did not significantly increase, suggesting that it is through post-transcriptional regulation. In the current version, through inhibition of protein synthesis with cycloheximide (CHX), we further demonstrated that the FASN protein was stabilized upon NS1 expression (Figure S10). The increased FASN protein expression may directly lead to the upregulation of lipid synthesis.

Furthermore, we showed that the N terminal domain of NS1, especially R38-K41 residues are essential for the binding as well as up-regulation of FASN function (Figure 4F&H, Figure S12&13).

7. It is not clear how the authors to determine the non-specific binding proteins during the IP step using protein G beads in the PED. It seems the authors only subtracted the GFP data.

Response: Yes, the reviewer is correct. We mainly used GFP-conjugated beads as a negative control and subtracted the non-specific binding proteins. Consistently, during AP-MS, we also used GFP as the negative control to calculate the MiST score. Moreover, we also performed 6 replicates using NS1 as the bait, and identified the consistently enriched ones as hits. The non-specific binding which randomly occurs during some of the replicates would be eliminated.

Minor concerns:

1. The correspondence email should be an academic email address.

Response: We were concerned that Du's email address at the medical school might be terminated after her graduation. Now we changed it to an UCLA email address.

2. Line 577. "binding experiments with NP". It should be NS1.

Response: We thank the reviewer for helping us correct this error. We changed it in the current version.

3. Fig.2B. It was confusing at the first look because the authors didn't indicate the non-specific band, which might be IgG light chain.

Response: Yes, the reviewer is correct, that is the IgG light chain. We now labeled it in the figure.

4. Line 207. Are the 115 proteins ISGs as they only appeared after IFN treatment?

Response: These factors identified only in the IFN treated condition are representative of three things. First, as the reviewer mentioned, some are indeed ISGs that are only expressed in the presence of IFN. Cross-referencing the list to the Interferon database (<http://www.interferome.org/interferome/home.jsp>) identifies 37 of the proteins as ISGs⁴. Second, some interactions will be specific to cellular changes that occur in the presence of IFN, but not be ISGs themselves. For example, changes in protein complex structure, localization, and cellular structure can alter interaction profiles independent of expression. Third, some of these interactions represent noise in the data. Even though all samples were run in a biological triplicate, bottom-up MS is not comprehensive and some proteins will be found in the IFN only condition by random chance.

Reviewer #2 (Remarks to the Author):

I have studied only the lipidomics part of this manuscript (as requested by the editor) and I have only one comment:

Line 271: "..media containing 50% U13C-glucose..". This is misleading to non-expert readers. Please say e.g. "...containing U13C-glucose/unlabeled glucose 1:1 (?? mg/ml)".

Response: We thank the reviewer for the suggestion. We made the following change: containing U13C labeled glucose:unlabeled glucose = 1:1 (glucose 2 g/L). The concentration of glucose is the same as commercial RPMI-1640 media.

Reviewer #3 (Remarks to the Author):

Du et al. established a novel protein protein-protein interaction screen termed Protein interaction detection by Exon Display (PED). The advantage of this method is the detection of low abundance binders (no bias towards unfavorable protein abundances in normal IP-based assays) and identification of specific domains. The authors evaluated the system by screening the interactome of the influenza virus NS1 protein. This revealed a large number of already described binding partners (but not all) and additional new ones. In functional studies the role of two factors were analyzed in more detail.

As expected from this group, this is a technical very sound study. The authors nicely describe and document the advantages and certain limitations of PED. The Enrichment scores in predicting the domain of a cellular protein with NS1 is not always obvious. The importance of the fatty acid synthetase (FASN) is not as obvious as the authors want.

Specific points:

1) Fig. 3E: Without knowing which domain of CPSF4 is interacting with NS1, the enrichment scores would not help very much. Is it possible to improve this?

Response: We thank the reviewer very much for this comment. As the reviewer suggested, we have turned down the point that we can accurately identify the binding domain using PED and pointed out the limitation of this approach for domain identification.

The expression and enrichment of individual exon and exon fragments theoretically allows for the identification of the binding domain of the prey. As evidenced in previous Figures 3E and 5F, data from this approach did lead to the enrichment of previously reported domains as well as new interfaces confirmed by deletion mapping. That being said, we agree with the reviewer's concerns that this approach may not be ideal for accurate de novo domain

identification due to a number of interfering factors. Firstly, not all binding sites are contiguous, so interactions and interacting domains won't always be faithfully represented by the library. Secondly, not all interfaces can be captured faithfully because a single exon or exon fragment may not fold or behave the same way as in the complete protein. Thirdly, while noise within the dataset can be partially counteracted by analysis of fragments across an entire gene, the level of noise of individual fragments can be high enough to potentially confound identification. Although we used a stringent selection criteria for enriched fragments (enrichment > 2 fold in all replicates), we still noted the false positives/ false negatives in the individual fragments. For example, the data in Figure S15 could indicate an enrichment in exons 23 and 33, which is not validated by the deletion mapping.

However, we think that PED can at least help us narrow down the possible binding domains of bait in some cases. Beside the evidence we showed in the current paper, we already have ongoing experiments using PED for the identification of drug targets among cellular proteins. We included an abstract and some key data from that project. Briefly, we used PED to identify RPL27A as the cellular target of Martynoside (MAR). MAR directly binds to RPL27A and attenuated 5-FU induced down-regulation of RPL27A by increasing protein stability. From the PED exon enrichment data, we observed that exons 4 and 5 of RPL27A may mediate the interaction. Using exon 4&5 as the binding pocket, we performed docking simulation and proposed the potential binding mode between MAR and RPL27A. Quantitative mass spectrometry showed that the addition of MAR reduced the ubiquitination on amino acid K92 and K94 of RPL27A, consistent with the predicted drug-docking model. Based on the above results, we further defined the functional mechanism of MAR binding to RPL27A, as details attached.

Beyond the scope of this paper, we are also working on improving the input library from random fragmented human exon library to a rationally designed synthesized oligo library. We anticipate that a more precisely controlled and less diverse input library will further increase the sensitivity for the identification of binding domains and reduce the noise.

2) Fig. 4b: The authors mention that FASN is required for viral growth, however, inhibition of viral replication is only 4-fold (despite favorable presentation of the results) and only tested in one cell line. It would be nice to substantiate this finding by using more appropriate cell lines (A549...).

Response: We thank the reviewer for the comments, and we agree that only using C75 and using transient shRNA transfection was not adequate to reveal the functional impact of

FASN for influenza viral replication. To further evaluate this issue, here we included two more drugs inhibiting FASN function, as well as direct knock-down of FASN gene expression using two different shRNA constructs. In all three drugs, we observed a ~5 fold reduction in viral TCID50 with no cell toxicity. We detected a 10-25 fold reduction at 36h post infection in A549 cells with stable knock down of FASN (Figure 4C, 4D).

3) Fig. 4F and G: Based on the enrichment scores it is unclear why fragment 3 (and to a certain degree fragment 5) is not binding to CPSF1. Thus, the enrichment score is a poor prediction of possible interacting domains (see point 1).

Response: We agree with the reviewer that this current approach may not be ideal for accurate de novo domain identification, although it may help narrow-down the potential binding pockets (response to point 1). We included a more nuanced discussion of the use of this approach for domain identification. We also included data from a separate project, to show that PED can help predict the binding pocket of a drug (as attached).

Reviewer #4:

This manuscript by Du and coworkers reported the use of one round of mRNA display selection coupled with high throughput DNA sequencing to identify potential human intracellular protein domains that bind to the influenza A virus (IAS) NS1 or its mutant, from a genomic human DNA fragment library with a size of 300-700 bp. 25 proteins were enriched from multiple replicates, including four previously reported NS1-binders (CPSF4, CPSF1, PABPC1, and NFX1). They performed co-IP studies of 10 selected proteins and found 8 appeared to directly interact with NS1, including three previously unreported NS1-binders (PSMB10, AKT1, and FASN). Preliminary biochemical analyses were performed to examine the possible roles of FASN/NS1 and CPSF1/NS1(D92Y) in the viral host interactions. In general, the work is interesting, but the major claims were not supported by the data provided in the manuscript.

1. One of the major challenges in the generation of exon library is how to get nucleic acid sequences that do not contain intron regions, since unremoved introns could be used for coding by the protein synthesis machinery, introducing artificial peptide motifs that do not exist in natural proteome. Whereas this is not a problem by using natural cDNA libraries derived from matured mRNAs in which introns are absent, the use of

genomic DNA as the starting material might have this issue. The ‘human exon library’ used in this work was generated by random fragmentation of genomic DNA and filtered to a size of 300-700 bp, followed by exon enrichment. The major concern is whether this is a real exon library or just a protein fragment library in which exons might contain a small fragment of peptide motifs coded by introns? Using the exon 3 of CPSF4 that the authors found to be the highest enriched segments in CPSF4 (Fig. 3E) as an example:

According to the genomic structure of CPSF4 from accession# EF191081.1, the CDS of CPSF4 are joined by the following regions:

2161..2263,7867..7917,11199..11351,13354..13449,13780..13873,15446..15518,17044..17214,19510..19578. The exon 3 is coded by 153 bps (from 11199 to 11351). Unless there is an effective mechanism to distinguish and remove the intron sequences, the translation will continue at the end of the last codon of exon 3, resulting in translation of the intron (11352 to 13353) between exon 3 and exon 4 until a stop codon is encountered, resulting in the following amino acid sequence (exon 3 underlined):

GMCPFRHISGEKTVVCKHWLRGLCKKGDQCEFLHEYDMTKMPECYFYSKFGK
APGALEALLRTRVQRGPLAAQCPHCLCLLFPFYSQRRTLAEPASLSQLLGDSV
AIFCSSLFFFFFF* This is very different from the natural exon 3 (underlined)-exon4
(italicized) fragment with the following amino acid
sequence:GMCPFRHISGEKTVVCKHWLRGLCKKGDQCEFLHEYDMTKMPECYF
YSKFGECNKECPFLHIDPESKIKDCPWYDRGFCKH

The problem is that the bolded polypeptide sequence is not only artificial and absent in the natural CPSF4, but also its presence at the C-terminus of exon 3 could alter the properties of exon 3 polypeptide, including its binding affinity, specificity, and mode with other proteins. It’s well known that the properties of polypeptides could be drastically changed due to the presence of several more residues at the N- or C-terminus. It is possible that the authors were lucky that the CPSF4 exon 3 sequences they isolated from the library still bound to NS1. Nevertheless, the binding properties of numerous other exon sequences could be altered due to the presence of the N- or C-terminal artificial motifs derived from introns, resulting in artifacts that are very hard to interpret. With that said, it is unclear whether the hybridization exon enrichment step that the authored reported in the protocol allowed the complete removal of the exon sequences that contain part of the introns? If yes, detailed explanation is needed, and related data should be provided.

Response: We truly thank the reviewer for the detailed comments, and we apologize for the lack of clarity in the first version. The reviewer is correct in stating that the hybridization exon enrichment step did not completely remove the introns. However, we provided the following precautions to minimize the influence of introns. This information is also included in the current version:

Firstly, as we performed random fragmentation of the genome, each exon was covered by many different fragments. Each fragment contained different flanking region for N- and/or C- terminus, but with the same exon region (line 174-175). Secondly, the intron regions had high possibilities to contain stop codons, encoding peptides that are largely unfolded or aggregated. These fragments were cleared during the FLAG purification step and be excluded from the input and output library (line 175-176). Lastly, during data analysis, we calculate the enrichment score with the number of sequences reads that overlap with the exon, regardless of its flanking intron region (Figure S5B, line 177-178).

We note that as the reviewer stated above, some of the intron regions would still be translated and some of them might affect the properties/binding ability of exon sequences, but we considered that with the diverse sequences that contain the same exon region, the effect of flanking regions was leveraged out and gave us a true signal.

Here, we used CPSF exon 3 from one input and one output library as a concrete example to further demonstrate this point. As the reviewer mentioned above, the exon 3 amino acid sequence should be:

GGMCPFRHISGEKTVVCKHWLRGLCKKGDQCEFLHEYDMTKMPECYFYSKF

In the input library, the read depth was 59M. Among these reads, we got 29 sequences containing exon 3, corresponding to a frequency of 0.49 RPM (reads per million). Each sequence comes with a different flanking region. In one of the output libraries, we detected 18 sequences covering exon 3 out of 20M reads, with a frequency of 0.89 RPM. Thus, the enrichment score for CPSF4 exon 3 is $0.89/0.49 = 1.82$. A schematic plot is provided as below:

(Frequency of Exon_i = No. of reads that overlap with Exon_i / Total Reads)

We also listed three sequences as examples below. In the input library, we detected the following three fragments, the bolded sequences belong to exon 3, while the rest is intron flanking region:

From Chr7 99045722-99046030 (hg19 reference genome), which encode:

PVPILPAGGMCPFRHISGEKTVVCKHWLRGLCKKGDQCEFLHEYDMTKMPECY
FYSKFGKAPGALEALLRTRVQRGPLAAQCPHCLCLLFPFYSQRRTLAEP

From Chr7 99045719-99045861, which encode:

HPVPILPAGGMCPFRHISGEKTVVCKHWLRGLCKKGDQCEFLHEYDM

From chr7:99045752-99046100, which encode:

CPFRHISGEKTVVCKHWLRGLCKKGDQCEFLHEYDMTKMPECYFYSKFGKAP
GALEALLRTRVQRGPLAAQCPHCLCLLFPFYSQRRTLAEPASLSQLLGDSVAIFCSS
LFFFFF

In the output library post selection, the first sequence was read twice, and the second was read once. However, we failed to detect the third fragment in the output library, probably due to the improper folded 3' flanking region (multiple F in the C terminus).

We are improving the input library for a better signal-to-noise ratio. We are currently designing and chemically synthesizing an oligonucleotide library that covers the whole human proteome. We anticipate that a more precisely designed library could provide us a cleaner result.

2. People in the field hypothesized that the use of less biased library with more evenly distributed input library could facilitate the identification of interactors with low abundance as suggested by previous studies, including those using mRNA display approach. However, a number of parameters could affect the identification of low abundance interactors, including the copy number of the interactor genes, the binding affinity, specificity, mode, and binding conditions. It is not clear whether the three novel NS1 binders identified by the authors can be classified as low abundance genes. More importantly, binding affinity and specificity data should be provided to further understand the putative interactions.

Response: We thank the reviewer for the comment. We also agree that various different parameters would affect the identification of binders, especially low abundant ones. To further prove the point that PED may facilitate low abundance binders, we were able to express the majority of the hits (23 out of 25) in 293T cells, and performed Co-IP to examine the binding with NS1 protein. As shown in the new Figure 2B, 14 binders were able to co-precipitate NS1 with various strength. Based on this result, we further compared the cellular abundance of identified binders in our matched PED and AP-MS datasets. The AP-MS datasets were independently obtained by Nevan Krogan's lab at UCSF. Consistent with our previous observation, the PED-identified interactors have a significant lower abundance in both cellular proteome dataset and RNA-seq data, when compared with interactors identified using AP-MS (Figure 3D).

Binding affinity measurements in vitro require purification of proteins and may not present the same physiological conditions for protein binding in cells. Thus, we did not plan to perform pair-wise binding affinity measurement for the identified binders. However, based on the strength of pull down in co-IP and western blot, we roughly categorized them into strong (CPSF4 and binders with similar strength, including CPSF1, NXF1, PSMB10 and TFAP2C); medium (AKT1, DDX6, PABPC1, FASN, SF3B2 and TTLL6), and weak binders (DNAAF3, INPP5B and USP54). All the 5 previously reported factors were categorized into strong or medium (CPSF1, CPSF4, PABPC1, NXF1 and SF3B2) binders. In the 9 newly identified factors, PSMB10 and TFAP2C are presented as strong binders, while AKT1, DDX6, FASN, TTLL6 as medium level binders (Table S1, Figure 2B).

For FASN which we studied in detail, we provided additional truncation mapping and mutation information to demonstrate the binding specificity. We showed that the N terminal domain of NS1 is the major contributor to FASN binding, and the double mutation in this domain, R38A/K41A, nearly completely disrupts the binding. Functionally, R38A/K41A

showed reduced ability to induce FASN up-regulation. Furthermore, it has reduced capacity to induce fatty acid synthesis as quantified by GC-MS.

3. Compared to AP-MS, much less interactors were identified, despite the dataset could be extended based on the results from other methods. In this reviewer's opinion, the strength of this paper is not the method, but the novel NS1 binders that were not reported previously and the related biological significance. Only 10 selected proteins were tested for co-IP. Why not all 25 proteins so that more novel and hopefully physiologically significant interactors could be revealed? Pull-down assay and binding affinity analyses can be easily performed for that number of interactors.

Response: We thank the reviewer for the suggestion. The number of identified hits largely depends on the selection criteria. To avoid false positive hits, we used a relative stringent threshold, which resulted in a small number of interactors. The current selection criteria provided us a decent overlapping rate with literature (5 out of the 25 hits were previously reported). We were also able to validate a large proportion of them by co-IP. As we stated above, in the current revised version, we were able to express the majority of the hits (23 out of 25) in 293T cells, and performed Co-IP to examine the binding with NS1 protein. Shown in Figure 2B, 14 of 23 binders were able to co-precipitate NS1 with various strength.

We also provided more detailed characterization on the importance of FASN on viral replication, including endogenous IP, generating stable knock-down of FASN in A549 cells using shRNA, and using multiple inhibitors. We believe that the revised version has a clearer demonstration of the functional impact of FASN, which we identified through the PED method.

4. Some advantages claimed by the authors are not unique to the method described in this manuscript. Tagging the target proteins is widely used in many proteomic searches. Current mass spectrometry allows for the identification of natural proteins with very low abundance. Lack of tissue type information is actually a disadvantage. Single round of selection coupled with HTS can also be applied to the mRNA display using natural cDNA library, and the biased problem could be addressed by different approaches.

Response: We thank the reviewer for the comments and we understand the reviewer's concern. We also agree that some of the improvements provided in PED can be extended to other methods/libraries, such as cDNA libraries. However, to our knowledge, our

presentation is the first to combine these improvements together, and to demonstrate the utility of this improved method in identifying protein-protein interactions in a large scale.

As we described in the Introduction and Discussion sections, tagging target protein with high-affinity epitopes is indeed widely used to overcome the limitation of high-affinity antibodies in proteomic searches. However, it also comes with limitations: it might be difficult to express tagged proteins in the cell type of interest; it may result in non-physiological levels of expression, and it is difficult to determine how the tag impacts the protein-protein interactions.

The sensitivity and accuracy of AP-MS continues to increase and we are also actively participating in it. We are very fortunate that Nevan Krogan, a leader in proteomics using mass spectrometry, performed the AP/MS part of our study. However, even with the most advanced 4D technique with timsTOF pro machine, the number of quantified proteins is limited to ~5000-7000. The detection of a protein is dependent on the biophysical properties of the peptides after digestion. The sensitivity, reproducibility and throughput are still incomparable to DNA deep sequencing. Thus, by converting the task of detecting a large number of different proteins to that of detecting nuclear acid sequences, we think that PED presents a method complementary to MS.

In the manuscript we mentioned that we have generated a cDNA library for mRNA display from A549 cells (line 139-149). We actually first tried single round enrichment coupled with HTS with this cDNA library, however, we had a hard time obtaining a high-quality hit list, with only ~3% overlapping rate with literature reported cellular factors.

As the cDNA library is generated by random fragmenting the mRNA extracted from A549 cells, we tried two ways of analyzing data. The first way is to perform peak calling similar as ChIP-Seq, and to examine the enriched peaks comparing input and output library. The second way is to split the whole cell transcriptome into 300 bp sliding windows and to examine which window (corresponding to which gene) has enrichment. Unfortunately, neither of the two methods gave us a good result. Even the robust NS1 binder, CPSF4, did not show up as an enriched cellular binder. As shown below with an IGV track plot, we did not see any enriched peak in CPSF4 gene with NS1 pull-down, compared with GFP.

The failure of using the cDNA library to identify NS1 binder was one of the major reasons that we switched to the human exon library. We cannot be sure why the exon library gave a better result. As the reviewer mentioned, there could be many reasons: copy number of the interactor genes, the binding affinity, specificity and mode, etc. However, as we kept all of the experimental conditions the same, we think the even distribution of input library is at least one of the main reasons that we obtained a better result.

References:

1. Hale, B. G., Randall, R. E., Ortín, J. & Jackson, D. The multifunctional NS1 protein of influenza A viruses. *J. Gen. Virol.* **89**, 2359–76 (2008).
2. Kochs, G., Garcia-Sastre, A. & Martinez-Sobrido, L. Multiple Anti-Interferon Actions of the Influenza A Virus NS1 Protein. *J. Virol.* **81**, 7011–7021 (2007).
3. Kuo, R. L. *et al.* Interactome analysis of the NS1 protein encoded by influenza A H1N1 virus reveals a positive regulatory role of host protein PRP19 in viral replication. *J. Proteome Res.* **15**, 1639–1648 (2016).
4. Rusinova, I. *et al.* INTERFEROME v2.0: an updated database of annotated interferon-regulated genes. *Nucleic Acids Res.* **41**, D1040–D1046 (2012).

Information on a separate study for reviewers and editors only

[Redacted]

Reviewers' Comments:

Reviewer #3:

Remarks to the Author:

The authors nicely addressed the open issues, however in Fig. 4D the viral titers in the control lanes differ by 100 fold! Is this reflecting the possible range of viral titers observed in independent experiments? If yes, all infection experiments +/- inhibitor have to be repeated simultaneously in biological triplicates. Why is the effect of chemical inhibition and knockdown of FASN on viral replication so different?

Reviewer #4:

Remarks to the Author:

In this revised manuscript, the authors provided many new data supporting their finding and characterization of novel NS1-binding proteins using mRNA display approach. The manuscript is much improved and the new data addressed some of this reviewer's concerns.

For question #1, the reviewer agrees that the enrichment scores are pretty good. The real concern is the nature of the exons thus generated. In an exon library, it is acceptable that only partial of an exon is covered in a given sequence, or some exons are not included in the library. However, it will be problematic if the exons or exon-fragments covered in the library contain many residues that do not exist in the proteome of interest. As this reviewer indicated previously, the tertiary structure and consequently the biological activity of most short peptides are sensitive to the flanking sequences. Using Chr7 99045722-99046030 (read twice in the output library post selection as explained by the authors), although it covers the full-length exon 3, 52 out of 103 amino acids are from the artificial intron coding. Folding this 103-residue polypeptide sequence using bioinformatic tools (such as I-TASSER) will most likely result in a 3D structure that is different from the native conformation. Typically, long artificial polypeptides are not soluble and get precipitated. This is, however, not the case in the mRNA-displayed peptides, because the long mRNA attached is extremely hydrophilic making most peptides soluble in the aqueous solution. The fact that the mRNA-displayed exons contain many unnatural residues, as confirmed by the authors, making these exon-containing polypeptides artificial and likely with tertiary structures different from exons only sequences. With that said, the reviewer agreed that CPSF is a real NS1-binding protein and the method described in this manuscript is still very useful. However, it could be limited to interactions that only require short peptides, or to highly stable protein motifs/domains whose structures are not easily disrupted by flanking artificial sequences. Therefore, this reviewer suggests to change the term of PED to more accurately describe the nature of and discuss the major limitations of this method.

For question #3, the authors used co-IP to demonstrate that 14 binders were co-precipitated with NS1 protein. While this result suggests these 14 proteins are potential NS1 binders, the experiment was performed by overexpressing the proteins of interest. Whether they interact with NS1 under physiological concentrations remains elusive, but can be addressed by pull-down and western assays using individual antibodies.

We would like to thank the editors and reviewers for their great questions and suggestions on our study. We have been thorough in our attempts to address all the points raised by the editors and reviewers. Additional evidences were also included to support the endogenous interaction between NS1 and FASN. We have revised 2 figure panels, added 2 figure panels and removed 1 supplementary figure. The main text is revised accordingly. Our point-by-point responses to the reviewers' concerns are provided in the following:

Reviewer: 1

The authors nicely addressed the open issues, however in Fig. 4D the viral titers in the control lanes differ by 100 fold! Is this reflecting the possible range of viral titers observed in independent experiments? If yes, all infection experiments +/- inhibitor have to be repeated simultaneously in biological triplicates. Why is the effect of chemical inhibition and knockdown of FASN on viral replication so different?

Response:

We thank the reviewer for the question. Indeed, the experiments presented for C75 were done using a different batch of virus and by different person (which were originally presented in the very first version - NCOMMS-19-08520- without the other two inhibitors). We have now provided the data for +/- FASN inhibitors that repeated simultaneously in biological triplicates (updated Figure 4D).

Currently we think the impact of chemical inhibition and knockdown of FASN on viral replication is consistent. The effect of inhibitor treatment on viral replication were examined at 24h post infection (Figure legend), which showed ~5 fold decrease at highest concentration. Similar impact on viral replication were also observed in KD cells at 24h post infection.

Reviewer #4 (Remarks to the Author):

In this revised manuscript, the authors provided many new data supporting their finding and characterization of novel NS1-binding proteins using mRNA display approach. The manuscript is much improved and the new data addressed some of this reviewer's concerns. For question #1, the reviewer agrees that the enrichment scores are pretty good. The real concern is the nature of the exons thus generated. In an exon library, it is acceptable that only partial of an exon is covered in a given sequence, or some exons are not included in the library. However, it will be problematic if the exons or exon-fragments covered in the library contain many residues that do not exist in the proteome of interest. As this reviewer indicated previously, the tertiary structure and consequently the biological activity of most short peptides are sensitive to the flanking sequences. Using Chr7 99045722-99046030 (read twice in the output library post selection as explained by the authors), although it covers the full-length exon 3, 52 out of 103 amino acids are from the artificial intron coding. Folding this 103-residue polypeptide sequence using bioinformatic tools (such as I-TASSER) will most likely result in a 3D structure that is different from the native conformation. Typically, long artificial polypeptides are not soluble and get precipitated. This is, however, not the case in the mRNA-displayed peptides, because the long mRNA attached is extremely hydrophilic making most peptides soluble in the aqueous solution. The fact that the mRNA-displayed exons contain many unnatural residues, as confirmed by the authors, making these exon-containing polypeptides artificial and likely with tertiary structures different from exons only sequences. With that said, the reviewer agreed that CPSF is a real NS1-binding protein and the method described in this manuscript is still very useful. However, it could be

limited to interactions that only require short peptides, or to highly stable protein motifs/domains whose structures are not easily disrupted by flanking artificial sequences.

Therefore, this reviewer suggests to change the term of PED to more accurately describe the nature of and discuss the major limitations of this method.

Response:

We really thank the reviewer for the suggestion. We have provided more detailed discussion on the limitations of our method (line 158-159, line 375-399 with changes highlighted). As the reviewer suggested, we have pointed out that the translated protein fragments might contain flanking regions of adjacent introns. These flanking regions may affect the tertiary structures of real exons, and making the selected hits biased toward interactions that only require short peptides, or to highly stable protein motifs/domains whose structures are not easily disrupted by flanking sequences. Furthermore, we have removed the “exon” features from the name of our method, to avoid any potential confusion. Instead, we re-name it as mRNA display with library of even-distribution (md-LED), reflecting that it has a more even representation across the proteome. The changes have been made throughout the entire manuscript.

Editor:

A revised manuscript would also need to provide quantification of repeat experiments to demonstrate that FASN protein is stabilized upon NS1 expression (current Figure S10) or remove this preliminary result and all associated claims.

Response:

We thank the editor for the comments. We have performed the experiment two times, thus we decided to remove the result (Supplementary Figure 10) and all associated claims in the revised manuscript.

Additional Responses:

In addition to the questions raised by the reviewers and editors, we have added two figure panels (Figure 4G and 4H, line 294-296) to further support that WT NS1 and FASN interact endogenously in A549 cells. Using stable A549 cells lines expressing WT NS1, mutant NS1 (R38A-K41A), NP and GFP, we observed that WT NS1 interact with endogenous FASN while R38A-K41A abolish the binding. Furthermore, this interaction is not dependent on RNA, as RNase treatment did not affect the binding. Two other RNA binding influenza viral proteins (NP and PA) were used as control, again revealing that the interaction between NS1 and FASN is not through non-specific RNA binding.